# ADAR1 RNA editing regulates endothelial cell functions via the MDA-5 RNA sensing signaling pathway

Xinfeng Guo[1,*], Silvia Liu[2,3,*], Rose Yan[1], Vy Nguyen[1], Mazen Zenati[1], Timothy R Billiar[1], Qingde Wang[1,3,4]

The RNA-sensing signaling pathway has been well studied as an essential antiviral mechanism of innate immunity. However, its role in non-infected cells is yet to be thoroughly characterized. Here, we demonstrated that the RNA sensing signaling pathway also reacts to the endogenous cellular RNAs in endothelial cells (ECs), and this reaction is regulated by the RNA-editing enzyme ADAR1. Cellular RNA sequencing analysis showed that EC RNAs endure extensive RNA editing, especially in the RNA transcripts of short interspersed nuclear elements. The EC-specific deletion of ADAR1 dramatically reduced the editing level on short interspersed nuclear element RNAs, resulting in newborn death in mice with damage evident in multiple organs. Genome-wide gene expression analysis revealed a prominent innate immune activation with a dramatically elevated expression of interferon-stimulated genes. However, blocking the RNA sensing signaling pathway by deletion of the cellular RNA receptor MDA-5 prevented interferon-stimulated gene expression and rescued the newborn mice from death. This evidence demonstrated that the RNA-editing/RNA-sensing signaling pathway dramatically modulates EC function, representing a novel molecular mechanism for the regulation of EC functions.

## Introduction

The cytosolic RNA-sensing signaling pathway is an important antiviral innate immune mechanism that responds to invading viral RNAs, leading to type I IFN and interferon stimulated gene (ISG) expressions (1, 2). Deregulation of this signaling pathway may result in autoimmune diseases if it reacts to cellular self RNAs (3, 4). It has been found that deficiency of the RNA-editing enzyme ADAR1 activates this innate immune signaling pathway and enhances ISG expression in mouse embryos (5, 6). RNA editing, specifically the A-to-I RNA editing catalyzed by ADAR1, acts as an essential regulator in innate immune homeostasis. However, the role of this innate

immune signaling pathway in non-infected cells for a specific cellular function has not been well characterized. As a posttranscriptional process, ADAR1-catalyzed RNA editing converts adenosine to inosine in cellular RNA transcripts at the regions with double-stranded structures (7, 8). It changes protein codons or splicing sites in mRNA, alters sequences within regulating small RNAs, and modifies the structure of dsRNA (9, 10). Emerging studies have shown that ADAR1 and RNA editing play essential roles in different cell types of cardiovascular tissues (11, 12, 13, 14), including cardiomyocytes (15, 16), vascular smooth muscle cells (12), and endothelial cells (ECs) (13). However, the mechanism of RNA editing in these cells has not been well defined.

ADAR1 has been shown to edit the transcripts for smooth muscle myosin heavy chain and α-actin to suppress their expression in vascular smooth muscle cells in phenotypic modulation and vascular remodeling in a mouse model (12). ADAR1's function in ECs was found to be associated with editing on specific RNA substrates such as cathepsin S (CTSS) (12) and miRNA487b (17). It was reported that ADAR1 carries out RNA editing in the 3′ UTRs of CTSS mRNA, regulating the RNA stability and expression level of this angiogenesis- and atherosclerosis-associated protease (13). ADAR1 levels and the extent of CTSS RNA editing are associated with changes in cathepsin S levels in patients with atherosclerotic vascular diseases, including subclinical atherosclerosis, coronary artery disease, aortic aneurysms, and advanced carotid atherosclerotic disease (13). These studies on cardiovascular cells suggest that a change in coded protein expression level, such as α-actin or CTSS, which are not known to be involved with immune regulation, is responsible for the effect of ADAR1 RNA editing. However, ADAR1 contributes to the editing at more than 100 million A-to-I editing sites, and most of them are identified in noncoding regions (18, 19), meaning that it is unlikely ADAR1 function is limited to some specific proteins. We and others have demonstrated that the major mechanism of ADAR1 function is to regulate innate immune activations (5, 6, 20, 21, 22). Nevertheless, whether the innate immune signaling pathway regulated by RNA editing plays a role in the cardiovascular cells and the underlying molecular mechanism has not been determined. In this study, through a series of analyses on

[1]Department of Surgery, University of Pittsburgh School of Medicine, Pittsburgh, PA, USA  [2]Department of Pathology, University of Pittsburgh School of Medicine, Pittsburgh, PA, USA  [3]Pittsburgh Liver Research Center, University of Pittsburgh Medical Center (UPMC) and University of Pittsburgh School of Medicine, Pittsburgh, PA, USA  [4]VA Pittsburgh Health System, Pittsburgh, PA, USA

Correspondence: wangqd@pitt.edu; billiartr@upmc.edu
*Xinfeng Guo and Silvia Liu contributed equally to this work.

variant KO mouse models and genome wide RNA sequence analysis, we demonstrated that the cytosolic RNA-sensing signaling pathway plays a critical role in ECs, and activation of this pathway is regulated by the endogenous RNAs edited by ADAR1. Specific deletion of ADAR1 in ECs causes neonatal mouse death with innate immune activation and immense EC ISG expression. The cellular function of ECs was dramatically impaired after ADAR1 deletion. Genome-wide RNA-seq analysis revealed that EC cellular RNA undergoes extensive RNA editing by ADAR1, especially in the RNA transcripts of short interspersed nuclear elements (SINEs). We found insufficient editing in these SINE transcripts in ADAR1-deficient ECs with RNA-sensing signaling pathway activation. Moreover, we demonstrated that blocking the RNA-sensing pathway via genetic deletion of the cellular RNA receptor MDA-5 reversed the ISG expression and completely rescued the mice from death. Without MDA-5 in ECs, the insufficiently edited cellular RNAs can be tolerated in mice. These findings demonstrated that the MDA-5 initiated RNA-sensing signaling pathway of innate immunity plays a critical role in the regulation of EC functions.

## Results

### EC-specific deletion of ADAR1 leads to postnatal death in mice

To delineate the specific function of ADAR1 in ECs and to determine the underlying molecular mechanism, we generated an EC-specific ADAR1 KO mouse model by crossing *VE-Cadherin-Cre* (*Chd5-Cre*) (23) and floxed *ADAR1* mice. Specifically, the VEN7 colony of *Chd5-Cre* transgenic mice was used in this study, as Cre activity in this colony has been reported to be more specific to ECs, including embryonic ECs, than other available Cre mouse lines (23). From F1 mice, we selected *ADAR1 wt/Lox; Cre⁺* genotype to breed with *ADAR1 Lox/Lox; Cre⁻* to produce the *ADAR1* homozygous and *Chd5-Cre⁺* mice for analysis. The presence of the *Cre* transgene and the heterozygous or homozygous floxed *ADAR1* gene was identified within the progenies using a PCR approach (Fig 1A and B). The genotype of *ADAR1 Lox/Lox; Cdh5-Cre⁺* was selected as the EC-specific *ADAR1* KO mice, referred to as ADAR1^EC-KO mice, whereas the littermates with *Lox/Lox; Cre⁻* and *ADAR1 wt/Lox; Cre⁺* genotype were used as controls. In contrast to germline *ADAR1* KO mice, which die in the embryonic stage at E11.5-12.0 d (6, 24, 25), *ADAR1 Lox/Lox; Cdh5-Cre⁺* mice were born alive. However, 75% of these *ADAR1 Lox/Lox; Cdh5-Cre⁺* pups died within 3 wk (Fig 1C). Obvious cyanotic signs were present after birth in some pups and became more evident before death (Fig 1D). Genotypic analysis showed that only the *ADAR1 Lox/Lox; Cdh5-Cre⁺* mice died early. Growth retardation was also evident in the second or third week of life, with the most severely affected is only one-third the size of their littermates (Fig 1E). Those *ADAR1 Lox/Lox; Cdh5-Cre⁺* mice surviving beyond 3 wk quickly caught up and achieved body sizes comparable to their littermates by 8–10 wk. The adult male and female ADAR1^EC-KO mice were fertile.

To determine why some *ADAR1 Lox/Lox; Cdh5-Cre⁺* pups survived beyond 3 wk while others perished, we assessed *ADAR1* deletion

efficiency among *ADAR1 Lox/Lox; Cdh5-Cre⁺* pups. Although the activity of *Chd5-Cre* was reported to start as early as E7.5 d during embryogenesis and progress to nearly complete penetration by E14.5 d (23), we found that only a partial deletion of *ADAR1* gene was achieved in *ADAR1 Lox/Lox; Cdh5-Cre⁺* pups, and the degree of *ADAR1* gene deletion in different mice varied significantly, even in pups from the same litters. As tested by semi-quantitative PCR, the deletion rates ranged from 20 to 70% in isolated lung ECs (Fig 1F). Thus, even partial deletion of ADAR1 in ECs led to postnatal lethality in mice. Furthermore, we found that *ADAR1* gene deletion was not detectable in ECs isolated from the *ADAR1 Lox/Lox; Cdh5-Cre⁺* mice survived beyond 3 wk (Fig 1G), indicating the Cre-mediated *ADAR1* gene recombination was inactivated in these mice. Survival of these mice was associated with the inactivation of the *Cre* transgene rather than tolerance of the absence of ADAR1.

It is known that ADAR1 deletion in hematopoietic cells leads to mouse death (24, 25, 26, 27), and nonspecific expression of *Chd5-Cre* may occur in some hematopoietic cells as previously reported (23). To clarify whether the *ADAR1 Lox/Lox; Cdh5-Cre⁺* newborn death was due to *ADAR1* gene deletion in hematopoietic and other cell types, we characterized *ADAR1* gene deletion in various tissues and cell types to confirm the specificity of ADAR1 deletion in ECs. ECs from lung tissue with >95% purity (Fig S1), together with DNA samples from other tissues, were analyzed for *ADAR1* gene deletion relative to undeleted allele. The signal for deletion was observed in ECs as tested in 2 wk-old *ADAR1 Lox/Lox; Cdh5-Cre⁺* pups; and no apparent ADAR1 deletion was observed in any non-EC DNA samples, including bone marrow cells (Fig 1H), whereas the deletion signaling was detected by the PCR with specific primers for the deleted allele in the highly vascularized organs (Fig S7). These findings support the conclusion that the detrimental phenotypes of this mouse model were due to ADAR1 deficiency caused functional defect of ECs.

### Deletion of ADAR1 in ECs causes multiple organ injury in ADAR1^EC-KO mice

To determine the organ-specific impact of ADAR1 deletion, gross and microscopic analyses were performed perimortem and postmortem in ADAR1^EC-KO mice. Consistent with the cyanotic phenotype, the most notable observed abnormality in the lungs was severe atelectasis involving either one or both lungs in all ADAR1^EC-KO mice that died within 3 wks of birth. In the most severe case, one side of the entire lung had completely collapsed (Figs 2A and S2). We often observed serous pleural fluid accumulated in the pleural cavities of mice with lung collapse, whereas in the control mice, pleural fluid accumulation was not observed. Partial atelectasis was observed in the ADAR1^EC-KO pups even not showing stress (Fig S2). HE staining of lung sections showed that many alveoli were collapsed and that the total air space was dramatically reduced (Fig 2B and C). The livers of deceased mice exhibited obvious pale patches on the surface, indicating patchy necrosis. Liver tissue sections showed areas with injured hepatocytes and cell death interspersed in otherwise relatively normal areas (Fig 2C). Less frequently, edematous small intestine was observed in some of the mice, with pathology studies showing shortened and scarce villi (Fig 2C). Although the kidney did not demonstrate gross abnormalities, tissue sections

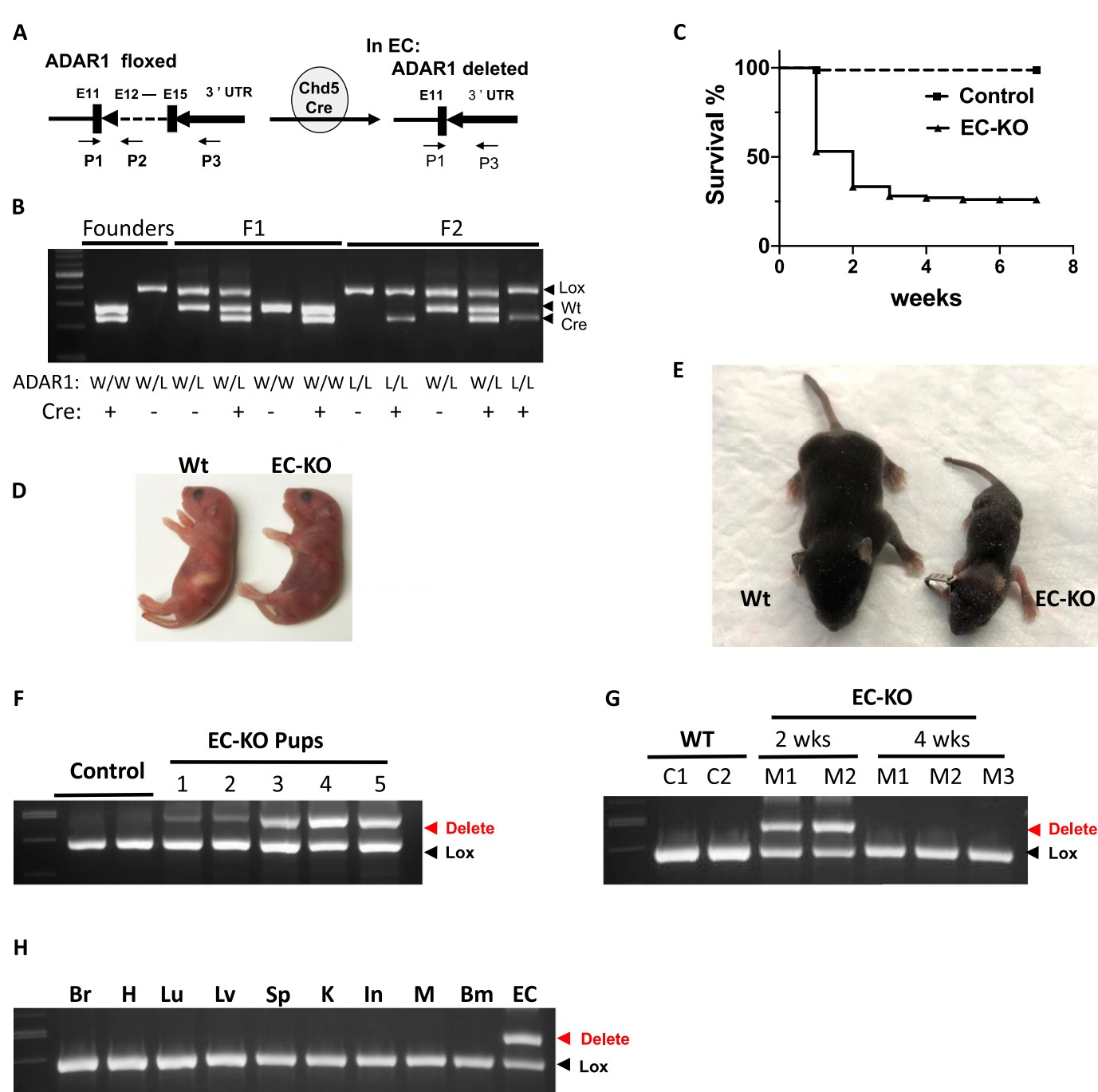

**Figure 1. Endothelial cell(EC)–specific deletion of ADAR1 causes postnatal death in mice.**
**(A)** EC-specific deletion of the *ADAR1* gene at exon 12–15 is mediated by *Cre* recombinase, which is driven by cadherin 5 promoter (VE-cadherin). The VEN7 transgenic mouse colony was used in this study. P1, P2, and P3 primers used for genotype analysis are indicated by the arrows. The PCR with mixed primer of these three was used to determine the relative quantity of floxed and deleted *ADAR1* alleles. **(B)** Typical genotyping analysis of founder, F1, and F2 progenies is shown in panel (B). **(C)** Half of the *EC-KO* pups (ADAR1 *Lox/Lox; Cdh5-Cre*⁺) died within 1 wk after birth, and about 75% died within 3 wk. **(D)** Cyanotic signs developed in some of the newborn pups, which became severe before they died. Shown here is an EC-KO pup with its *wild type* littermate at day 2 after birth. **(E)** Growth was severely retarded in some of the EC-KO pups, which usually cannot survive. **(E)** Shown in panel (E) is one EC-KO pup with its littermate at two and half week after birth. **(F)** PCR analysis of *ADAR1* gene deletion in ECs isolated from 1 wk old EC-KO pups shows dramatic variation in different mice. **(G)** ECs isolated from EC-KO mice that survived to 4 wk of age were analyzed for *ADAR1* gene deletion by PCR and compared with EC-KO mice at 2 wk of age. No obvious deletion was observed in 4-wk-old mice. **(H)** *ADAR1* gene deletion in brain, heart (H), lung (Lu), liver (Lv), kidney (K), intestine (In), muscle (M), bone marrow (Bm), and lung ECs was analyzed; *ADAR1* deletion was shown to only have occurred in ECs.

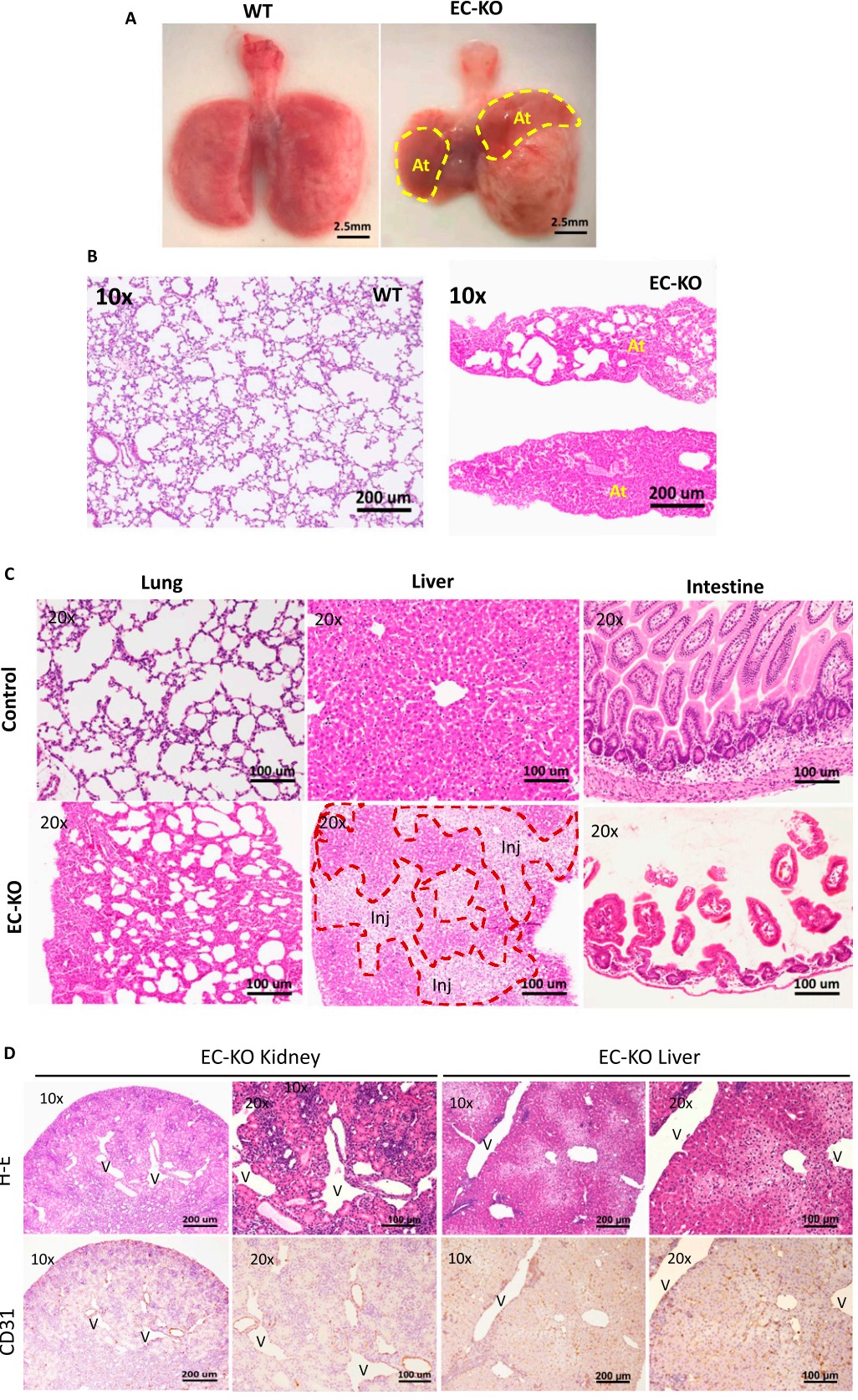

revealed disrupted structures with dramatically dilated blood vessels. The cortical thickness was decreased, and the medullar structure was distorted by empty spaces. We used EC marker CD31 antibody to stain the tissue sections and found that ECs were stained by CD31 antibody lining the enlarged tubular structures, confirming the spaces were dilated or enlarged blood vessels (Fig 2D). This vascular engorgement was also observed in liver tissues (Fig 2D). In contrast, morphologic changes in the brain, heart, spleen, muscle, and skin were not as dramatic as those seen in the lung, kidney, and liver.

### Innate immune activation in ADAR1-deficient ECs

To determine the molecular basis that caused detrimental tissue injuries, we investigated potential signaling pathway activations in ADAR1-deficient ECs. Previous studies showed that *ADAR1* gene deletion causes activation of the IFN-signaling pathway and elevation of ISG expression (5, 6, 28). We sought to determine whether ADAR1 deletion in ECs also leads to activation of the IFN-signaling pathway. A panel of ISGs was assessed with real-time RT-PCR in primary ECs isolated from the lungs of ADAR1[EC-KO] newborns. Of the 23 ISGs in the panel, 16 genes were significantly up-regulated up to 1,000-fold, including *Rsad-2, CCL-5, Ifit-3b, Ifit-1,* and *Mx-2* (Fig 3A). As described above, only a portion of the EC population isolated from ADAR1[EC-KO] mice was ADAR1-deficient. To confirm that the elevated ISG expression was specifically attributable to *ADAR1* deletion, we need an EC population in which the ADAR1 gene is efficiently deleted. We developed an in vitro method for the efficient deletion of the *ADAR1* gene in cultured ECs. First, we isolated ECs from an inducible ADAR1 KO (i-KO) mouse model described previously (22, 28, 29, 30, 31, 32, 33). Cells from i-KO mice can be induced to delete the *ADAR1* gene in vitro by adding tamoxifen (TM) to the culture medium (22, 28). We confirmed that more than 90% *ADAR1* gene was deleted in cultured i-KO ECs 48 h after TM treatment. The RNAs from the TM treated and non-treated ECs were analyzed for ISG expression (Fig 3B). We selected the seven most highly expressed ISGs to be tested for their expression levels in these i-KO ECs. The increased ISG levels in the TM treated i-KO ECs were consistent with the results from the ECs isolated from EC[ADAR1-KO] mice, and *Rsad-2, CCL-5, Ifit-3b, Ifit-1,* and *Mx-2* remained the most highly expressed genes (Fig 3B).

For a complete gene expression profile analysis, we carried out a genome-wide RNA-seq and bioinformatics analysis on EC RNAs. Two i-KO mice from independent litters were used for EC isolation, and cells from these two mice were induced with TM to delete the *ADAR1* gene. The RNAs of these treated cells, together with RNAs of the control non-induced ECs (CON) were analyzed in parallel

(Fig 4A). Efficient *ADAR1* gene deletion of exons 12–15 was confirmed by RNA read alignment (Fig 4B). Genome-wide gene expression profiles were analyzed across the four libraries. As shown in Fig 4C, principal component analysis showed that cells from two independent mice were clearly grouped by control and TM treatment. Differential expression analysis was then applied to the CON and TM groups, and the levels of 365 genes were significantly different (differentially expressed gene, DEG) in TM-induced ECs versus control ECs, including 236 up-regulated and 129 down-regulated genes (Fig 4D and Table S1). We further performed Ingenuity Pathway Analysis on the DEGs to detect pathways enriched by these genes. 49 pathways were detected using false discovery rate (FDR) = 5% (Table S2), and the top 20 are shown in Fig 4E. Notably, immune and cell death signaling pathways were the highest scored pathways, with the IFN-signaling pathway being the most significant among all the identified pathways. The expression patterns of DEGs involved in this pathway are illustrated in Fig 4F, showing that many ISGs were highly up-regulated.

### Cellular functions of ECs are impaired

Next, we proceeded to determine whether the activated IFN-signaling pathway in ECs affects a specific cellular function due to ADAR1 deficiency. Because *ADAR1* gene deletion in different ADAR1[EC-KO] mice varied significantly, for reproducible results from in vitro EC function assays, we again used the cultured primary ECs isolated from i-KO mice. After the ECs were isolated, we optimized the conditions for ADAR1 deletion, and found that induction with 400 nm TM for 48 h led to efficient *ADAR1* gene deletions (Fig 5A and B). These conditions were used to assess the impact of ADAR1 on EC function.

First, we observed that cell numbers were decreased with TM induction compared with non-induced cells, whereas we did not note a significant difference in cell morphology. To determine whether ADAR1 deletion affects the growth rate of the cells, we treated the cells with TM for 48 h and then replated the ECs in parallel with the non-treated cells with the same number of cells per plate. Total cell numbers were counted at 24-, 48-, and 72-h time points. Whereas non-treated cells demonstrated continual growth, the TM-treated cells exhibited nearly no expansion (Fig 5C). We stained the cells for the Ki-67 proliferation marker and found that whereas a large portion of primary ECs were positively stained, a much lower proportion of the TM-treated ECs were Ki-67–positive (Fig 5D and E). TM induction also significantly inhibited EC migration in the culture dishes coated with 0.1% gelatin, as shown by the dramatically smaller area covered by the migrating cells (Fig 5F and

**Figure 2. Multiple organ injuries in endothelial cell-KO mice.**
**(A)** Severe atelectasis occurred in all ADAR1[EC-KO] mice that died within 3 wk of birth. Panel (A) shows the postmortem lungs of an ADAR1[EC-KO] mouse and a wild-type mouse at two and a half weeks of age. The circles indicate the collapsed areas. The right lung almost completely collapsed. At, atelectasis area. **(B)** HE-stained lung sections of ADAR1[EC-KO] mice show many alveoli were collapsed, and total air space was dramatically reduced. "At," indicate the areas with complete atelectasis. **(C)** The microscopic analysis found apparent morphologic changes in multiple organs, including the lungs, liver, and intestine. On the liver sections, areas of injured and necrotic hepatocytes were scattered in the relatively normal hepatocyte areas. The injured and necrotic areas were circled and labeled with "Inj." On the small intestine sections, villi became scarce, and the length of the villi was dramatically decreased with structure interrupted. The scale bar is 100 $\mu$m. **(D)** On the H-E–stained sections of the kidney and liver, the tissue structures were disrupted by the tubule-like empty spaces. CD31 antibody staining showed the tubule spaces were lined by the endothelial cells with CD31 positive staining, confirming they were dilated-blood vessels. "V," indicates the enlarged vessel structures. Images were taken with 10× and 20× objective magnifications as indicated. Scale bars are 200 and 100 $\mu$m, respectively.

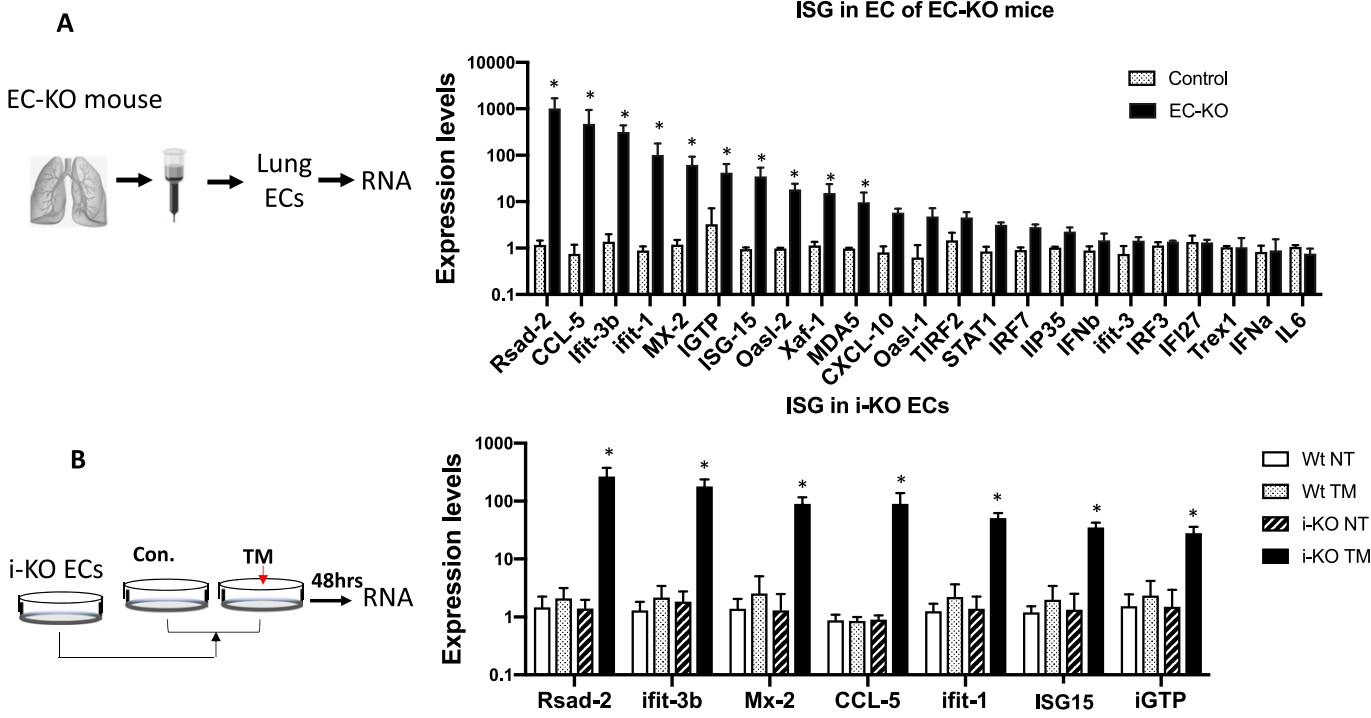

**Figure 3. Up-regulated interferon-stimulated gene (ISG) expression in ADAR1-deficient endothelial cells (ECs).**
**(A)** ECs isolated from ADAR1[EC-KO] mice were analyzed for ISG expression levels. Real-time RT-PCR was used to quantify a panel of 23 ISG expression activities. Significantly increased expression was observed in 16 genes in the cells of EC-KO mice compared with the littermate controls. $P < 0.05$, n = 3 (control) and 4–6 (EC-KO). **(B)** Expression of seven selected ISGs was also quantified in cultured ECs isolated from inducible ADAR1 KO (i-KO) mice. *ADAR1* gene deletion was induced by adding tamoxifen to the culture medium. Non-induced ECs (NT) and ECs isolated from *wild type* mice were used as controls. The expression of all these genes was significantly increased in tamoxifen-treated i-KO ECs. $P < 0.05$, n = 3 (control) and 3–4 (i-KO). RNA expression levels were determined using the ΔΔCt method with average of *GAPDH* and *HPRT* as endogenous control.

G). Finally, the potential impact of ADAR1 on the tube formation capacity of the ECs was assessed by culturing ECs on Matrigel. There was significantly less net tube formation and total tube length after TM induction compared with non-induced cells (Fig 5H and I). These findings show that ADAR1 deficiency imposes a dramatic effect on EC functions.

### EC RNA transcripts of repetitive elements were extensively edited by ADAR1

To delineate how ADAR1 regulates the IFN pathway, we examined changes in the substrate RNAs of ADAR1. ADAR1 is one of only two enzymes in mammalian cells that carry out A-to-I RNA editing (7, 8). We expected that ADAR1 deletion would dramatically decrease the cellular RNA editing level in its RNA substrates, and the reduced editing sites would be identified by comparing the RNA sequences of control and TM-treated ECs. Thus, to determine which RNA substrates were edited differently after ADAR1 deletion, we assessed the overall RNA editing status in four related EC RNA-seq libraries (Fig 4A). Because ADAR1 catalyzes adenosine (A) to inosine (I) and I is read as guanosine (G), the mismatches, including A/G mismatches (in positive-strand genes) and T/C mismatches (in negative-strand genes), were called from each of the four RNA libraries with reference to the mouse genome. To reduce the

influence of sequencing errors in identification of A-to-I editing sites, the following criteria were applied in our analysis: (1) total sequencing depth ≥ 5; (2) editing reads ≥ 3; and (3) editing rate ≥5% and ≤95%. Using these criteria, 11,955 and 9,475 potential editing sites were detected in the EC1 CON and TM libraries, respectively, and 10,624 and 12,210 sites in the EC2 libraries. Whereas some of these potential editing sites fell into the 5′ and 3′ distal regions (within 50 kbp from the gene region), the majority were distributed in gene regions, especially in the introns and 3′ UTRs, as shown by the result of EC1 CON RNA sample in Fig 6A and B. We analyzed the common features of these potential editing sites and found about half fell in region of repetitive elements (RRs), especially SINE, LTR, and LINE, as shown in Fig 6C and D with the EC1 CON RNA library results. Similar distribution patterns were also observed in the other three RNA libraries (Fig S3A–L).

To reduce the influence of SNPs on editing site identification, the editing rates of each editing site were compared between the two libraries of each mouse (i.e., CON and TM libraries of the same mouse were compared). All the editing sites fell into three groups: CON > TM, not significantly different, and CON < TM (defined in the Materials and Methods section). We considered only the sites with different editing rates as reliable editing sites in our analysis. There were 6,915 and 5,171 sites with higher editing levels in control cells (CON > TM) detected in the first and second mouse, respectively (Fig

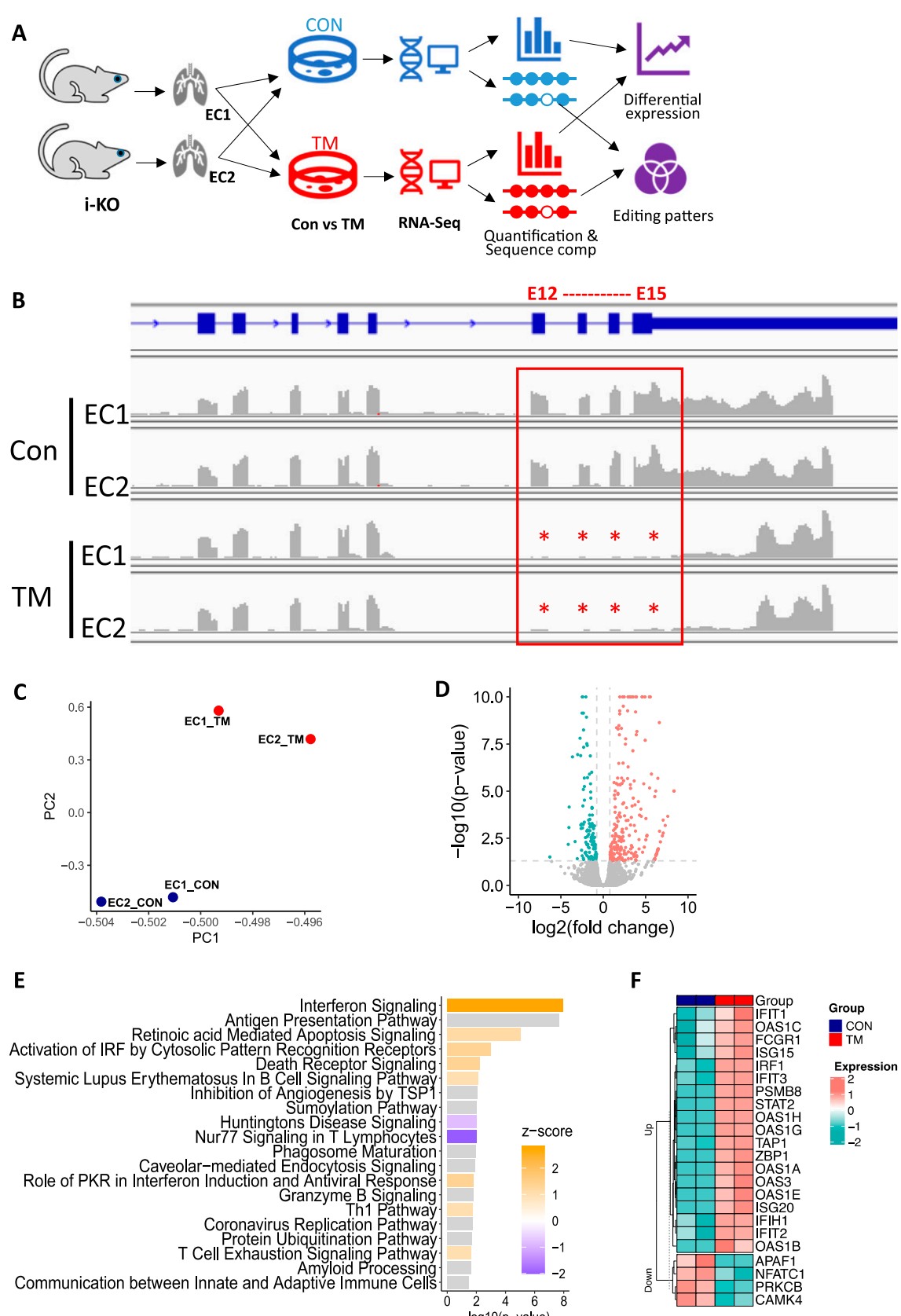

S4A–L), and among them, 1,773 sites were shared by these two mice (Fig 6E). In contrast, 4,539 and 5,533 sites were detected with the CON < TM editing rate in the first and second mouse (Fig S5A–L), and among these, 942 sites were shared (Fig 6F). Although most of the shared sites fell in gene regions, there were more sites in the 3′ UTR in the CON > TM group than in the CON < TM group (Fig 6G–J). However, two-thirds of the CON > TM editing sites fell in RRs in contrast to the CON < TM editing sites, of which only one-third fell in RRs (Fig 6K and L). Nevertheless, most of the editing sites in RRs located to SINEs in both groups compared with other RRs (Fig 6M and N).

### ADAR1 deletion in ECs reduces RNA editing in SINE RNA transcripts

Next, we focused our analysis on RNA transcripts annotated to RRs. In each mouse, the total editing sites in SINE, LINE, and LTR were counted in both CON and TM-treated ECs. Notably, the numbers of editing sites in these three major subgroups of RRs were well conserved between mouse 1 and mouse 2, and most of the editing sites fell in SINE (Fig 7A). In addition, the numbers of SINEs, LINEs, and LTRs that were subjected to editing were also very similar in these two mice, with SINE as the most frequently edited RR (Fig 7B). We calculated the average editing efficiency of each editing site in these subtypes of RR in the CON and TM-treated ECs. Remarkably, the average of editing efficiency in SINEs was reduced in both mice with TM treatment, whereas it was not observed in LINEs and TLRs (Fig 7C).

We further explored the editing levels of these RRs in CON and TM ECs, considering both the editing efficiency of each editing site and the total number of editing sites in each RR. If an RR contains more editing sites and the average editing rate was higher in control cells than in the corresponding TM-treated cells, we defined it as a high editing RR (CON > TM). On the contrary, a RR with less editing sites and a lower average editing rate was defined as a low editing RR (CON < TM). Using these definitions, we identified 2,188 and 1,417 high editing RRs in each mouse; among them, 467 RRs were shared by the ECs from the two mice (Fig 7D). Similarly, CON < TM RRs were also identified. As a summary, Fig 7E shows both the unique and common RRs for the CON > TM and CON < TM groups. More RRs were identified in the CON > TM group, and more common RRs were shared by two mice in this group, illustrating that high editing efficiencies in CON cells were more robust than in TM-treated cells. Furthermore, we mapped the common high and low editing RRs to three major categories: SINE, LTR, and LINE. As shown in Fig 7F, most of the high editing RRs were in SINEs, and there were significantly more high editing SINEs than low editing SINEs. In contrast, the number of high and low editing LTRs and LINEs were similar. Within the SINE subcategories, including B1, B2, B4, and others, the numbers of high editing RRs were dramatically higher than low editing RRs in all categories, and B1 showed the greatest difference between the high and low editing RRs (Fig 7G).

Among the high editing B1 RNA transcripts, multiple editing sites were often found in each B1 element (Table S3). These editing sites found by RNA-seq analysis were confirmed with three selected B1 SINE elements through Sanger sequencing experiments, including that in the intron of the Sppl2a gene (chr2: 126892815-126892950), 3′ UTR of the Nipa1 gene (chr7: 55977578-55977700), and 3′ UTR of the Slfn5 gene (chr11: 82962556-82962680). RNAs from wild type ECs were subjected to Sanger sequencing analysis after RT-PCR amplifications. Significant editing at multiple sites in these three B1 elements was successfully validated (Fig S6A–C). This result supports that ADAR1-catalyzed RNA editing potentially impacts the dsRNA structures formed by these repetitive SINE RNA transcripts (5, 6, 21, 34), thereby altering their immunoactivities in the innate immune response. The RNA-sensing signaling pathway of innate immunity was likely activated by RNA structure changes in the ECs of ADAR1$^{EC-KO}$ mice, causing severe organ injuries.

### Blocking the MDA-5–mediated innate immune pathway rescued ADAR1$^{EC-KO}$ mice from death

We next explored whether RNA-sensing signaling pathways were activated in the ECs of ADAR1$^{EC-KO}$ mice. It has been shown that the cellular RNA receptor MDA-5 mediates innate immune activation in ADAR1-deficient cells (5, 6, 21). We hypothesized that IFN pathway activation triggered by cellular RNAs in ADAR1 KO ECs is responsible for the newborn death in ADAR1$^{EC-KO}$ mice. To test this hypothesis, we deleted MDA-5 to block the RNA-sensing pathway in ADAR1$^{EC-KO}$ mice by crossing ADAR1$^{EC-KO}$ mice with MDA-5$^{-/-}$ mice (35) to generate a double KO mouse model with both MDA-5 and ADAR1 deleted within the ECs (ADAR1$^{EC-KO}$/MDA-5$^{-/-}$) (Fig 8A and B). We found that removal of MDA-5 completely rescued ADAR1$^{EC-KO}$ mice from death; all the double KO (d-KO) mice survived to adulthood without obvious abnormalities (Fig 8C). ADAR1 deletion in ECs were confirmed via PCR analysis (Fig 8D). Notably, more efficient ADAR1 deletion in ECs was observed in adult ADAR1$^{EC-KO}$/MDA-5$^{-/-}$ double KO mice than that observed in newborns of ADAR1$^{EC-KO}$ mice, indicating that the absence of MDA-5 renders mice tolerant to EC-ADAR1 deletion. These data demonstrate that the MDA-5–mediated signaling pathway accounts for pathological responses in ADAR1-deficient ECs.

### MDA-5 mediates innate immune activation in ADAR1-deficient ECs

To determine whether MDA-5 deficiency compensated RNA editing activity in ADAR1-deficient ECs, we examined whether RNA editing was altered in both MDA-5– and ADAR1-deficient ECs. Again, we used ADAR1-inducible KO ECs and crossed the ADAR1 i-KO mouse with the MDA-5$^{-/-}$ mouse (ADAR1 i-KO/MDA-5$^{-/-}$ double KO) to generate inducible double KO ECs. RNAs isolated from TM-induced ADAR1 i-KO, ADAR1 i-KO/MDA-5$^{-/-}$ double KO, and wild-type ECs

**Figure 4. Genome-wide differential gene expression in control and ADAR1-deficient endothelial cells (ECs) via RNA-seq analysis.**
**(A)** RNA analysis pipeline showing that tamoxifen (TM)-treated and non-treated ECs from two independent ADAR1 i-KO mice were used for analysis. **(B)** ADAR1 deletion of exon 12–15 in TM-treated ECs was confirmed by RNA read alignment. **(C)** Principal component analysis shows that cells from two independent mice were grouped by cell treatment. Blue: control samples; red: TM-treated samples. **(D)** Gene expression analysis illustrated by volcano plot showing the differentially expressed genes. Red: up-regulated genes; green: down-regulated genes. **(E)** Top significant pathways detected by Ingenuity Pathway Analysis based on the differentially expressed genes. Positive Z-score (orange) indicates activation, and negative score (blue) represents inhibition. **(F)** Heat map for the differentially expressed genes involved in the interferon signaling pathway.

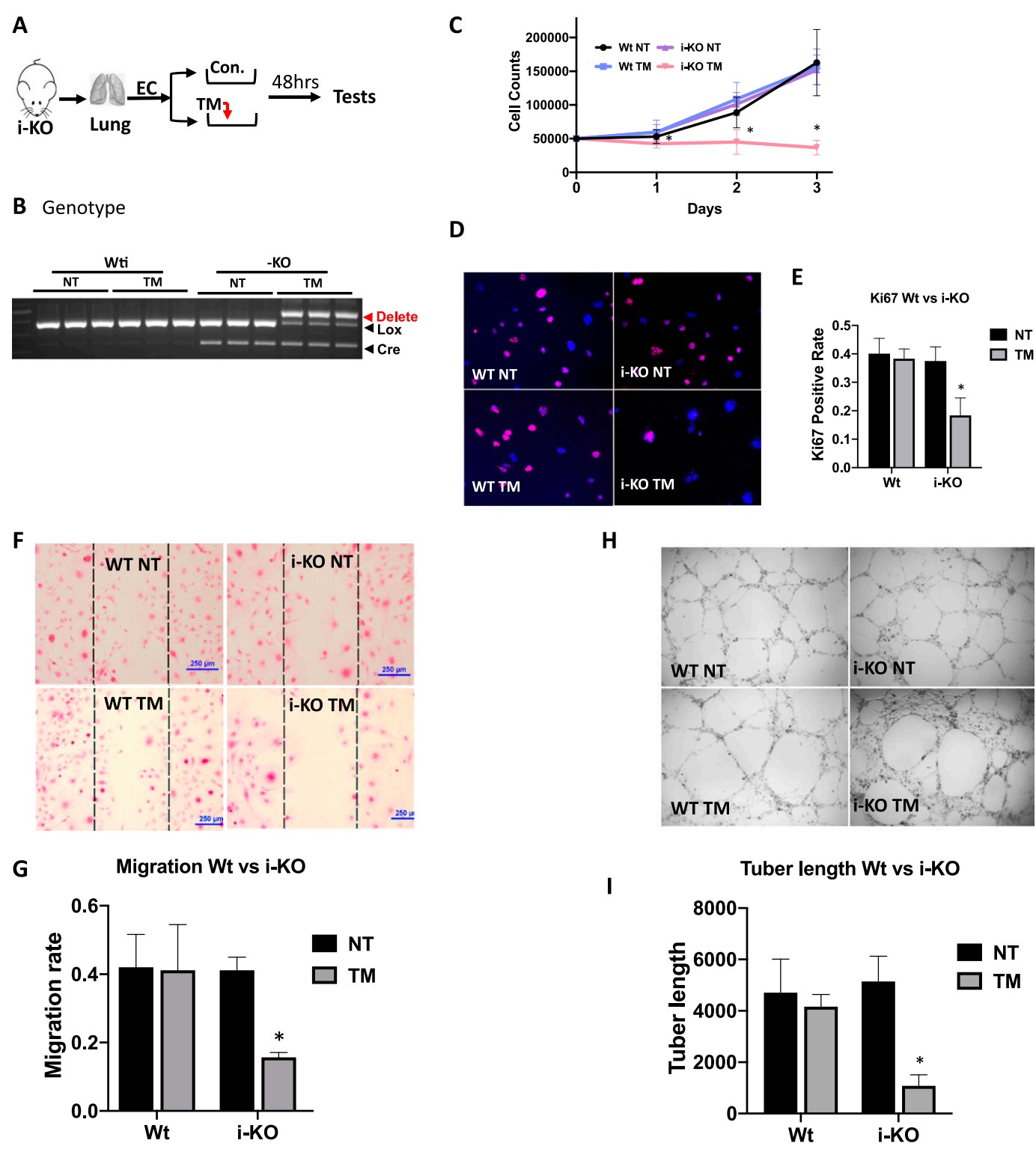

**Figure 5. Endothelial cell (EC) cellular function was impacted by ADAR1 deficiency.**
**(A)** ECs were isolated from the lungs of 1- to 2-wk-old inducible ADAR1 KO (i-KO) mice and cultured in vitro for cellular function assays. Tamoxifen (TM) was added to culture medium for 48 h to induce *ADAR1* gene deletion before assays were performed, as shown in panel (A). **(B)** *ADAR1* gene deletion in the tested ECs was confirmed by PCR analysis before EC function was assessed. Efficient *ADAR1* gene deletion in TM-induced i-KO ECs was shown by the ratios of the deleted and floxed gene PCR products. **(C)** Cell growth rate was monitored by counting the cell numbers at 24-, 48-, and 72-h time points after replating the same number of the TM-treated and non-treated ECs. TM induction dramatically reduced i-KO EC numbers compared with the controls. $P < 0.05$, n = 9. **(D, E)** Proliferating ECs were monitored by Ki-67 antibody staining. The number of positively stained TM-treated i-KO cells was significantly reduced compare with the control cells. $P < 0.05$, n = 9. **(F, G)** Cell migration rate was assessed by measuring the areas covered by cells that migrated, crossing the edges from both sides (between the lines). The cell migration areas were dramatically smaller in

were subjected to Sanger sequencing after RT-PCR amplification. As expected, editing on the B1 RNAs was decreased in TM-induced i-KO cells compared with the controls, and the editing level in the double i-KO ECs (i-d-KO) was the same as that in cells from the induced i-KO cells, as shown by the B1 RNA in the 3′ UTR of *Slfn5* mRNA (Fig 9A), indicating that reduced RNA editing remained in the rescued mice.

Then, we assessed ISG expression levels in the ECs of ADAR1^EC-KO^/*MDA-5*^−/−^ double KO mice. ISG expression in the ECs from double KO mice was significantly suppressed, similar to the levels in ECs from wild-type mice (Fig 9B). In addition, we also assessed the ISG levels in cultured ADAR1 i-KO ECs and ADAR1 i-KO/MDA-5 double KO (i-d-KO) ECs with TM induction. Consistent with the results from primary ECs from ADAR1^EC-KO^/MDA-5^−/−^ double KO mice, MDA-5 deletion also prevented ISG expression in cultured i-d-KO ECs (Fig 9C). *ADAR1* and *MDA-5* gene deletions in the tested cells were confirmed (Fig 9D), showing ADAR1 was deleted after TM induction in ECs in which MDA-5 was either present or deleted, whereas ISG expression was up-regulated only in ECs with MDA-5 present. These data indicated that innate immune activation in ADAR1^EC-KO^ ECs was mediated by MDA-5, and that this pathway is normally silent in the presence of ADAR1 in physiologic conditions.

# Discussion

In this series of studies intended to identify the roles of ADAR1 in ECs and the underlying mechanism that modulates EC functions, we show that innate immune homeostasis in ECs is critically regulated by ADAR1-catalyzed RNA editing. Extensive RNA editing, especially editing in SINE transcripts, is required for maintaining innate immune homeostasis that is essential for postnatal survival in mice. All the detrimental phenotypes of the EC-specific ADAR1 KO mice and the dramatically changed cytokine expression in ECs can be prevented by removal of cellular RNA receptor MDA-5; thus, we demonstrated that activation of the MDA-5–initiated RNA-sensing signaling pathway is an important mechanism for EC function regulations.

In this study, we choose to use Cdh5-Cre to excise the floxed *ADAR1* gene in ECs, as this Cre line was shown to be highly specific for ECs (23). The recombination activity of this Cre on a foxed report gene was reported to start from the embryonic stages in ECs with minimal nonspecific activity in embryonic liver hematopoietic cells (23). However, no obvious abnormality in embryonic development was observed in our study, and only a partial deletion of the *ADAR1* gene was detected in the ECs isolated from ADAR1^EC-KO^ newborn mice. The actual activity of Cre on floxed *ADAR1* gene was not as efficient as shown in the reporter gene (23). Therefore, we cannot exclude a potential effect of ADAR1 in embryonic development. However, we confirmed that apparent *ADAR1* gene deletion, as quantified relative to the undeleted floxed *ADAR1* alleles, was

restricted to ECs (Fig 1H). And the *Adar1* gene deletion in the tissue DNA samples could be detected by the PCR with the specific primer set for deleted allele, confirming the ADAR1 deletion in the organs (Fig S7). This ensured that the phenotype observed in this mouse model was attributable to the defect in EC functions. In addition, the activity of Cre in the postnatal EC-KO mice varied significantly in different pups, Fig 1F, explains the different viability of the EC-KO mice at neonatal ages. Only about half of the floxed ADAR1 gene was deleted in the most efficient EC-KO newborns, demonstrating that the neonatal mice do not tolerate even a partial ADAR1 deletion in the ECs. As a result, the highly vascularized organs, such as the lungs and kidneys, were most significantly affected (Fig 2). As shown by the in vitro assays, EC functions could be substantially impacted by ADAR1 deficiency (Fig 5). The EC is a major cell type of the blood-air barrier in the lungs; thus, the functional defect of the ECs significantly impaired the functions of the lungs. In the kidney and liver, we observed large empty spaces in the H-E tissue sections, which disrupted the typical structures of these organs. By CD31 antibody staining, we found that the CD31 positive ECs were lining the spaces, confirming that the spaces were dilated or enlarged blood vessels. This enlarged vessel structures indicated that ADAR1 deficiency in ECs may affect the angiogenesis or blood vessel remodeling in the kidney and livers. Likely, the cellular RNAs in ECs need to be processed by ADAR1 to modulate EC functions to support the newborn to adapt to the new environment. Without ADAR1, EC RNAs failed to be processed appropriately, activating MDA5-mediated RNA sensing and up-regulation of IFN signaling, leading to failures of the affected organs.

ADAR1 deletion in ECs resulted in dramatically increased expression of multiple ISGs, including *Ifit 1, CXCL10, ISG-15, IRG7, MDA-5,* and others, Fig 3, which were also observed in ADAR1-deficient embryos (5, 6, 21, 26). However, the levels of type I IFN, including *IFN-α* and *IFN-β*, were not increased in ECs, which is different from the ADAR1 deficient embryos (5, 21, 26) and fibroblast cells (22). IFN independent ISG expression may happen in certain viral infection conditions (36). In ADAR1 deficient ECs, the ISG expression might also be IFN independent. The IFN-α receptor 1 is essential for IFN signal transduction; however, deletion of its coding gene Ifnar1 in ADAR1 KO embryos did not rescue them to birth, although it extended the embryo survival time for a few days (6, 21), indicating the cellular defect caused by ADAR1 deficiency remains in Ifnar1^−/−^ cells. This evidence also supports that ADAR1 deficiency causes an IFN independent effect in the cells.

MDA-5 KO completely rescued the ADAR1EC-KO mice from postnatal lethality, Fig 8, and the rescued mice did not show a notable difference in their development and reproducibility. Previous studies showed that the deletion of MDA-5 completely rescued the editing deficient *Adar1*^E861A/E861A^ mutant mice (5). However, either deletion of MDA-5 or MAVS could not fully rescue the *Adar1*^Δ7−9^ KO mice (6), and MAVS deletion did not fully rescue the *Adar1*^Δ2−13^ KO mice (21). Instead, they rescued the KO embryos to term. Most of the rescued mice died within 1 d after birth, and none

---

TM-treated i-KO ECs than the controls. *P* < 0.05, n = 3 (control) and 9 (i-KO). **(H, I)** Tube formation capacity was tested on Matrigel. Far fewer tube structures developed in TM-treated i-KO ECs, and the total tube length was significantly less than the controls. *P* < 0.05, n = 3 (control) and 4–6 (i-KO).

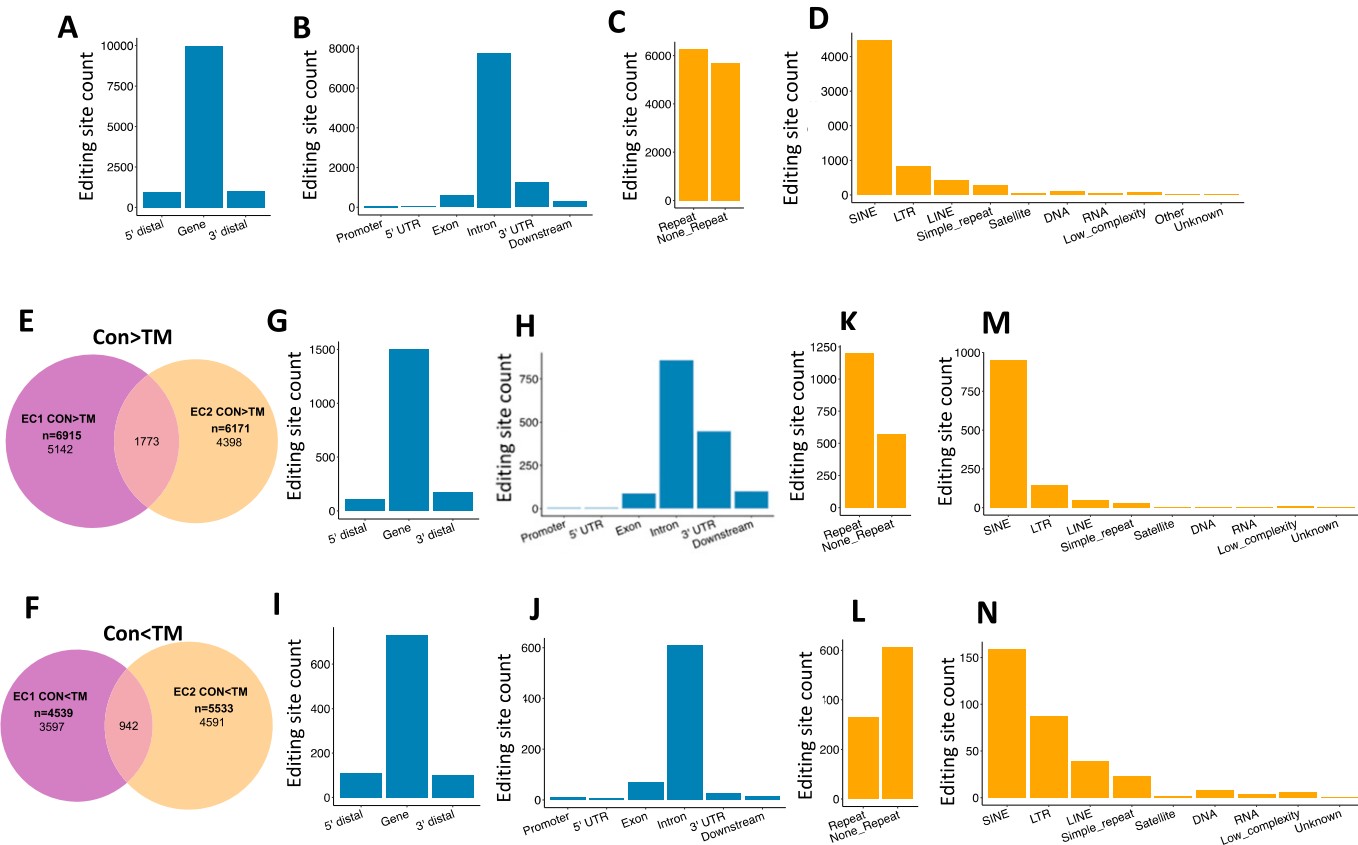

**Figure 6. Distribution of RNA editing sites in endothelial cells (ECs).**
Extensive RNA editing was identified in EC RNA transcripts by RNA sequence analysis. **(A, B)** show that most editing sites fell into gene regions, especially intron and 3′ untranslated regions. **(C, D)** show that about half of the editing sites were located in repetitive regions, with most of them falling in short interspersed nuclear element, long terminal repeat, and long interspersed nuclear element. **(A, B, C, D)** are the results of control (CON) ECs of the first mouse. **(E)** Venn diagram showing overlap between EC1 and EC2 for the CON > TM editing sites. **(F)** Venn diagram showing the overlap between EC1 and EC2 for the CON < TM editing sites. **(G, H)** Gene distribution of the common CON > TM editing sites in EC1 and EC2. **(I, J)** Gene distribution of the common CON < TM editing sites in EC1 and EC2. **(K, L)** Distribution of the EC1 and EC2 common editing sites with CON > TM and CON < TM, respectively, in repeat and non-repeat gene regions. **(M, N)** Distribution of the EC1 and EC2 common editing sites with CON > TM and CON < TM, respectively, in subcategories of repetitive regions. Panel (E, G, H, K, M) are the EC1 and EC2 common editing sites with CON > TM editing rate. Panel (F, I, J, L, N) are the EC1 and EC2 common editing sites with CON < TM editing rate.

of them survived beyond 10 d (6, 21). The KO of specific ADAR1 P150 was also embryonic lethal, but more than half of the $P150^{-/-}$ were rescued by MAVS deletion to 3 wk of age with multiple organs morphologically abnormal (6). $Adar1^{\Delta2-13}$ KO deleted the sequence coding most of the entire ADAR1 protein (21), including the Z-DNA/RNA binding domain, RNA binding motif, and the catalytic domain, and $Adar1^{\Delta7-9}$ KO affects both RNA binding and catalytic domains (6, 26). In our EC-KO model, the Exon 12–15 was floxed for deletion. We confirmed that the sequence of exon 12–15 was efficiently removed from RNA transcript in the ADAR1 deficient ECs, whereas transcription of other regions of the *ADAR1* gene was not affected (Fig 4B). We previously showed that the deletion of exon 12–15 in embryos did not lead to truncated protein in the cells (24, 37). Although we could not get an efficient deletion of ADAR in EC-KO mice for protein analysis, we analyzed the protein from in vitro cultured i-KO ECs by Western blot. No truncated protein was detected in the $Adar1^{\Delta12-15}$ KO ECs (Fig S8). Therefore, the complete rescue of ADAR1 EC-KO mice by MDA-5 deletion was not due to the potential truncated ADAR1 protein retaining the activity of RNA binding and Z-DNA/RNA binding. We did test whether $Adar1^{\Delta12-15}$ KO

embryo could be rescued by MDA-5 deletion. Same as the $Adar1^{\Delta2-13}$ and $Adar1^{\Delta7-9}$ KO, the double KO mice were born alive but could not survive postnatally (data not shown). Therefore, the rescue of ADAR1 EC-KO mice by MDA-5 deletion was due to EC-specific ADAR1 deletion. In ADAR1 EC-KO mice, ADAR1 deletion occurred in half or less of the ECs. It was significant enough to cause severe organ injuries; however, blocking the MDA-5 pathway prevented the ISG expression and restored the critical functions of the EC for supporting the mouse survival. Thus, the homeostasis of innate immunity regulated by ADAR1 is very critical for maintaining EC functions.

A previous study showed that ADAR1 edits on CTSS mRNAs which plays a critical role in the regulation of EC functions, and increased editing on CTSS was found associated with angiogenesis in atherosclerotic vascular diseases in humans (13). Edited CTSS mRNAs resistant to TUDOR-mediated RNA degradation, therefore promoting EC proliferation (13). This significant finding revealed a mechanism of ADAR1 editing in human ECs under pathologic conditions. However, we could not find significant CTSS editing in mouse ECs, most likely because editing in human CTSS mRNA occurs within the 3′ UTR, which encompasses an Alu repetitive

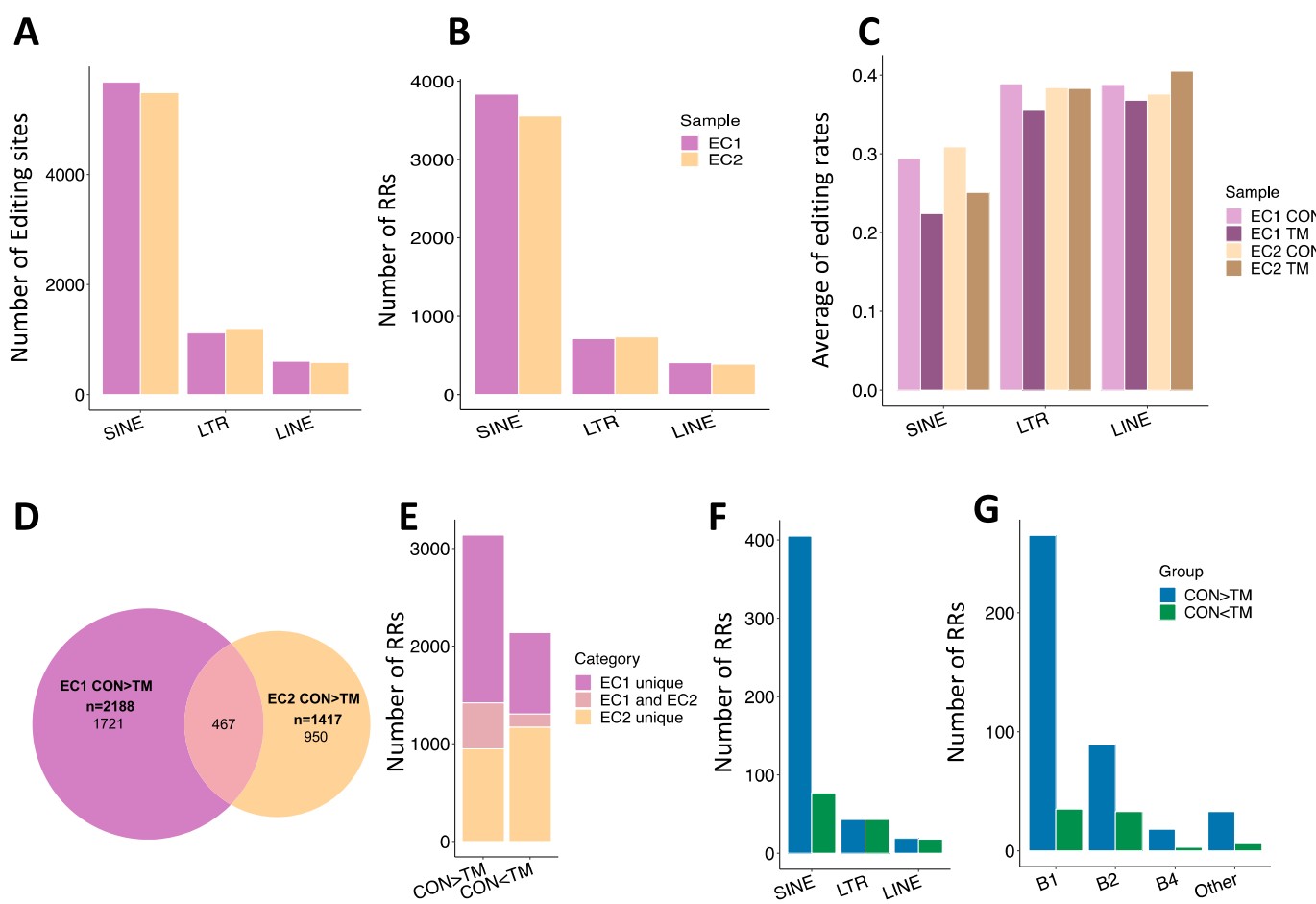

**Figure 7.  Decreased editing in short interspersed nuclear element (SINE) RNA transcripts.**
**(A)** Number of editing sites across SINE, long terminal repeat (LTR), and long interspersed nuclear element (LINE) regions. **(B)** Number of repetitive regions (RRs) across SINE, LTR, and LINE regions. **(C)** Average editing rate across SINE, LTR, and LINE regions. **(D)** Venn diagram showing the overlap between EC1 and EC2 for the CON > TM editing RRs. **(E)** Number of editing RRs with CON > TM and CON < TM detected in EC1 only, EC2 only, and shared by the two mice. **(F)** Distribution of common RRs of the two mice across SINE, LTR, and LINE. **(G)** Distribution of the common RRs of the two mice across subcategories of SINE.

element, and this Alu element does not exist in the mouse genome (34, 38). Nevertheless, most A-to-I RNA editing in mouse ECs occurs to the noncoding regions, predominantly in the repetitive elements, consistent with that found in humans (10, 34, 39). Whereas most editing occurs to Alu repetitive elements in humans (38, 40), most editing occurs to B repetitive elements in mice, the counterparts to Alu in humans (5, 34, 40). It was shown that Alu editing was associated with innate immune activation in human cells (41, 42). We speculate that ADAR1 RNA editing may also play a role in human ECs for innate immune homeostasis through RNA editing in addition to editing CTSS mRNA in ECs.

The molecular mechanism of ADAR1 in ECs was implicated through RNA editing on specific RNA substrates such as CTSS mRNA or microRNAs. In contrast, we demonstrated in this study that regulation on the RNA-sensing pathway is the key mechanism that underlies the function of ADAR1 in ECs. Most of ADAR1-mediated editing in ECs occurs at noncoding regions of RNAs in introns and 3′ UTRs on repetitive RNA elements, and especially in SINE transcripts, which form dsRNA structures. Deletion of ADAR1 in ECs leads to dramatically elevated ISG expression, and further deletion of the

cytosolic RNA receptor MDA-5 reverses ISG expression back to within normal range and completely recues ADAR1[EC-KO] mice from death. This finding strongly supports the presence of an ADAR1 RNA-editing/RNA-sensing signaling axis in ECs that regulates innate immune homeostasis and plays a critical role in the regulation of EC functions.

ECs play critical roles at multiple stages of inflammatory reactions (43, 44, 45). However, endogenous RNA-triggered activation of the innate immune signaling pathway in ECs has not been extensively studied. Here, we provide evidence that cellular RNA sensing through MDA-5 activates inflammatory signaling in ADAR1-deficient ECs, and that this disrupts EC function, leading to the early death of newborn mice. The complete protection of mice deficient in MDA5 from the pathological features of EC-specific ADR1 deletion confirms the central role of MDA5 in the detection of RNA for immune activation in ADAR1-deficient ECs. It is possible that inflammatory cytokines produced and released by ECs act on other cell types or synchronize with other inflammatory mechanisms in the pathologic development of inflammatory vascular diseases.

**A**  Double KO: ADAR1$^{EC-KO}$ + MDA-5$^{-/-}$

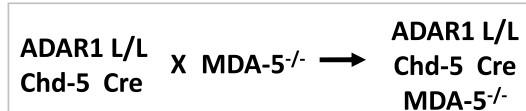

**B**  Double KO genotyping

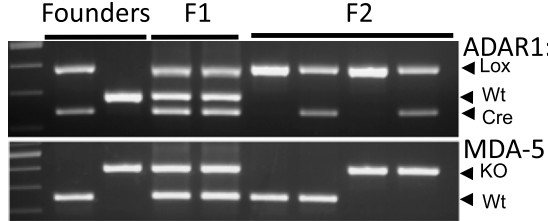

**C**  Double KO survival

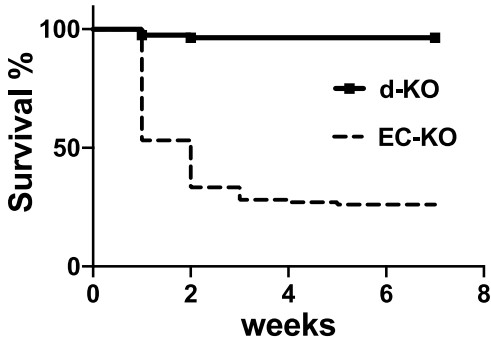

**D**  ADAR1 deletion in ECs of d-KO mice

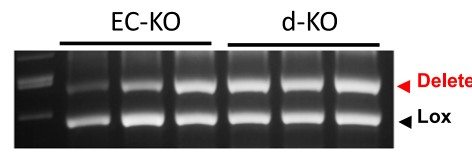

**Figure 8.  *MDA-5* deletion rescues ADAR1EC-KO mice from postnatal death.**
**(A)** ADAR1$^{EC-KO}$ mice were crossed with *MDA5*$^{-/-}$ mice and in the resulting double knockout (d-KO) mice, both *ADAR1* and *MDA-5* were deleted in ECs. **(B)** PCR genotyping was used to monitor the genetic status of floxed *ADAR1, Cad-5 Cre*, and *MDA-5* and identify the d-KO mice. Panel (B) shows the typical genotypes of the breeders, progenies of F1 and F2 generations. **(C)** The postnatal death that occurred to ADAR1$^{EC-KO}$ was not observed in the d-KO mice. The d-KO mice survived to adulthood without noticeable abnormality. **(D)** *ADAR1* gene deletion in the ECs isolated from 6-wk-old d-KO mice was tested using PCR analysis and compared with 2-wk-old ADAR1$^{EC-KO}$ mice. Efficient *ADAR1* deletion was observed in d-KO mice.

Although the MDA-5-mediated RNA-sensing signaling pathway was shown to underlie the detrimental phenotype of our ADAR1$^{EC-KO}$ mouse model, we also noticed that the elevated ISG expression was not completely abolished, Fig 9B and C, and the cellular functions of ADAR1-deficient ECs tested by in vitro assays were not completely restored to wild type levels by MDA-5 deletion Fig S9A–C, indicating that some of the EC defects caused by ADAR1 deletion are MDA-5 independent. Whereas MDA-5 seemed to be the major RNA receptor in ECs to sense the altered RNAs due to ADAR1 deficiency, other signaling pathways such as RIG-I or different molecular mechanisms may also respond to ADAR1 deficiency. We did not test whether MAVS KO completely rescue the EC functions in this study.

It is known that recoding activities of ADAR1 on mRNAs and microRNAs also regulate EC functions. RNA editing in protein coding regions may result in amino acid changes such as the Q/R site editing in filamin A and B mRNAs (46, 47). Filamin mRNAs, especially filamin A mRNAs, were reported to be highly edited in human and mouse cardiovascular tissues (48). Patient-derived RNA-seq data demonstrate a significant drop in FLNA editing associated with cardiovascular diseases, and blockage of FLNA editing in mice increased vascular contraction and diastolic hypertension (48). Although ADAR2 was reported to be responsible for most of the FLNA editing (46, 47), we found that editing on filamin A and B mRNAs was also dramatically decreased in ADAR1-deficient mouse ECs (Table S4). Therefore, editing in protein coding regions by ADAR1 likely also contributes to the regulation of EC functions. However, the complete rescue of survival and normal

phenotype of the ADAR1$^{EC-KO}$/MDA-5$^{-/-}$ double KO mice indicated that the recoding activity of ADAR1 on specific protein coding mRNAs is insignificant and loss of the specifically recoded RNAs can be tolerated by the rescued mice. Our RNA-seq analysis showed that most of the RNA editing occurred to SINE RNA transcripts in noncoding regions of mRNAs. Editing on these SINE RNAs is likely a critical regulator for cellular homeostasis. More studies will be necessary for a complete understanding of RNA editing in ECs, whereas the major function in regulation of the MDA-5–mediated RNA-sensing signaling pathway has been demonstrated in this study.

## Materials and Methods

### Animal model preparation

All animals used for this study were kept in a specific-pathogen-free animal facility and the protocol was approved by the University of Pittsburgh's Institutional Animal Care and Use Committee. Floxed ADAR1 mice were prepared as described previously (24, 28). VE-Cre mice (Chd-5 Cre), Stock # 006137, and *MDA-5*$^{-/-}$ mice, Stock #015812, were purchased from Jackson Laboratory. VE-Cre mice were first crossed with floxed ADAR1 to produce the EC-specific ADAR1 KO mouse strain. Then, ADAR1 EC KO mice were crossed with *MDA-5*$^{-/-}$ mice to produce ADAR1/MDA-5 double KO mice. PCR was used for genotypic analysis of all the mice used for this study. PCR

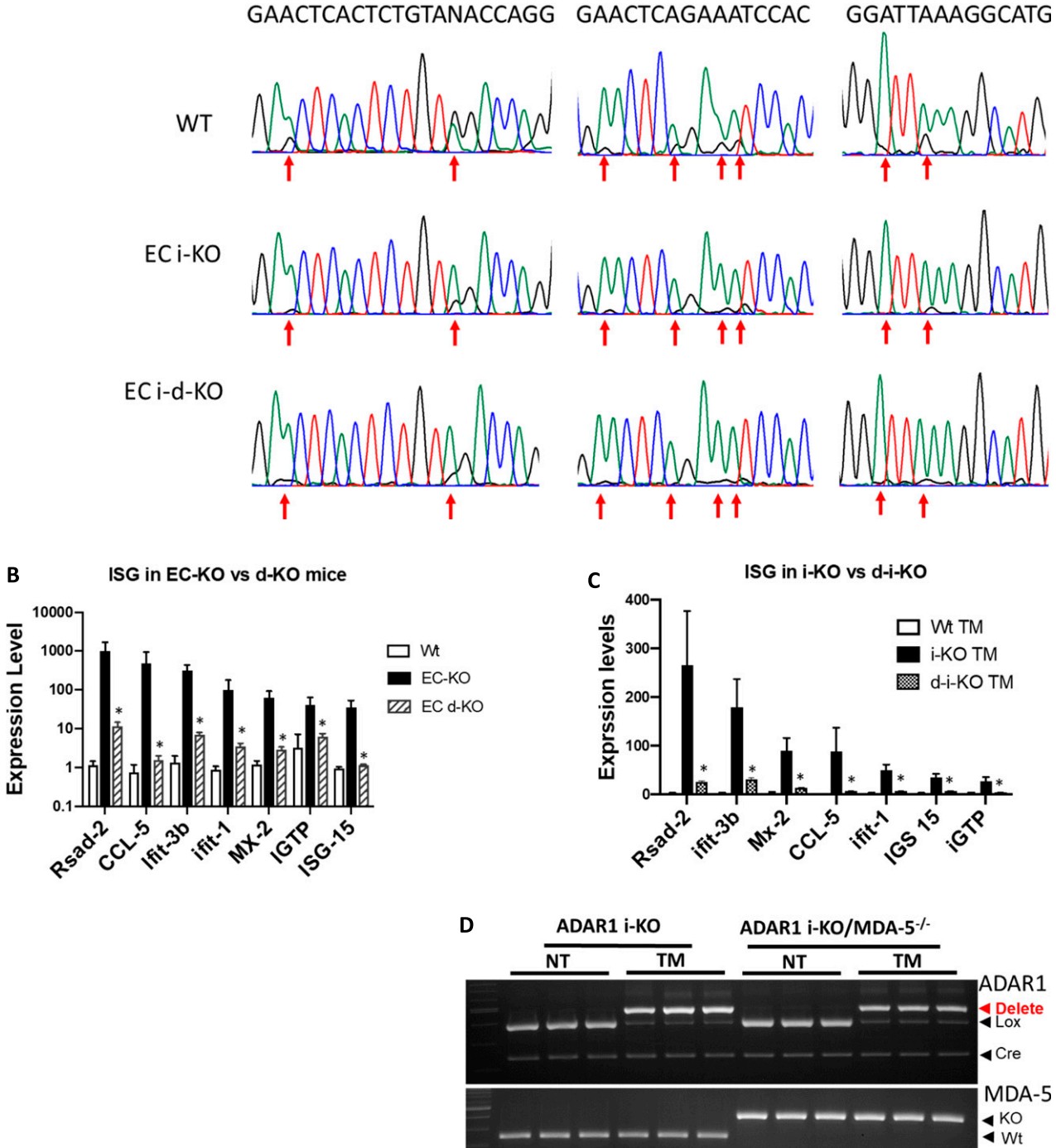

**A** RNA editing level not compensated d-KO mice

3' UTR of Slfn5

**Figure 9. MDA-5 deletion diminishes interferon-stimulated gene (ISG) expression in endothelial cells (ECs) with reduced short interspersed nuclear element (SINE) RNA editing caused by ADAR1 deficiency.**
**(A)** ECs isolated from *wild type (WT)*, inducible ADAR1 KO (i-KO), and inducible ADAR1/MDA-5 double knockout (i-d-KO) mice were cultured in vitro with tamoxifen (TM) induction for ADAR1 gene deletion. RNA editing in SINE RNA transcripts was assessed through RT-PCR and Sanger sequencing, and compared between wild-type, i-KO, and i-d-KO ECs. Panel (A) shows the chromatographs of Sanger sequencing result of EC RNAs. The multiple editing sites in the specific B1 SINE element located in the 3' untranslated region of the *Slfn5* gene were indicated by arrows. Obvious editing, as shown by the G peaks (black) together with A peak (green), presents in wild type EC RNAs. The editing levels (G peak areas) in i-d-KO ECs were same as the i-KO ECs, which were dramatically lower than the *WT* ECs. **(B)** ECs isolated from *WT*, EC-KO, and EC

conditions and primer sequences were as reported previously for ADAR1 (24, 28) and followed Jackson's protocol for MDA-5.

## Mouse phenotype analysis

Newborn mice were observed for phenotypic changes daily from birth. Newborn death was observed from the first day after birth. Samples from dead mice were collected for genotype analysis. Cyanotic signs were observed before death and the numbers of dead pups were recorded each day. The newborns of each litter were numbered for phenotype recording before genotype analysis was performed on mice between 1 and 2 wk old, and ear tags were used to identify the pups after 2 wk. The relative sizes of the newborns were recorded.

## Pathology study

Newborn mice were examined daily for general health changes from birth. Death frequently occurring after birth within the first 3 wk was observed and recorded. Postmortem examination was performed immediately after newborn death to identify gross morphological abnormalities. Perimortem examination was also performed on mice showing obvious cyanotic signs. Major organs and tissues were fixed with formalin overnight and embed in paraffin. Tissue sections were subjected to standard hematoxylin and eosin (HE) staining for morphologic studies.

## ISG expression in ADAR1 EC-KO lung ECs

Lung ECs were isolated from 1- to 2-wk-old ADAR1 EC-KO mice following an established protocol (49, 50). Dissected lungs were digested to single cells using collagenase I (Cat. no. LS004196; Worthington Biochemical Co.), 3 mg/ml and DNase I (Cat. no. DN25; Sigma-Aldrich), 10 $\mu$g/ml at 37°C for 45 min. ECs were first negatively selected with CD45 microbeads (MACs, Cat. no. 130-052-301) and then positively selected with CD31 microbeads (MACs, Cat. no. 130-097-418). Isolated ECs were directly subjected to RNA isolation with PureLink RNA Mini Kit (Cat. no. 12183018A; Invitrogen). 100 ng of total RNA was used for reverse transcription with iScript cDNA Synthesis Kit (Cat. no. 1708841; Bio-Rad) for 20 min. Real time PCR was performed with Bio-Rad kit (Cat. no. 170-8885) for gene expression with the CFX Connect Real-Time PCR Detection System. RNA expression levels were determined using the $\Delta\Delta$Ct method with average of *GAPDH* and *HPRT* as endogenous control, as described previously (22, 28).

## Inducible ADAR1 gene deletion in cultured ECs

ADAR1-inducible KO mice, prepared as described in our previous studies (22, 28, 31, 32, 33), were used for primary EC preparation. Single lung cells were prepared with collagenase digestion as described above. Then the cells were first subjected to magnetic bead sorting with CD31 antibody (Cat. no. 553370; BD Bioscience) and

Dynabeads (Cat. no. 11035; Thermo Fisher Scientific) according to the manufacturer's instructions. Then the cells from each mouse were seeded to two 6-cm dishes and cultured in EC growth medium: DMEM with 20% fetal bovine serum, 0.6% EC growth supplement (ECCS, Cat. no. 212-GS; Cell Application Inc.), and 100 $\mu$g/ml heparin in complete DMEM at 37°C, 5% $CO_2$ for 3–4 d until the cells reached 95% confluency. Then cells were lifted with trypsin and subjected to second selection with anti ICAM-2 (CD102) antibody (Cat. no. 553326; BD Bioscience). Isolated ECs were stained with anti CD31 antibody to monitor the purities. 400 nM 4-OH-TM (Cat. no. 579002; Sigma-Aldrich) was added to cultured ECs at passage three and culture was continued for 48 h to induce ADAR1 gene deletion. ADAR1 gene deletion was confirmed using PCR analysis. TM-treated cells and untreated control cells were used for cell function analysis, gene expression analysis, and RNA-seq studies.

## EC proliferation and migration assays

Primary i-KO ECs were replated to 3-cm dishes in triplicates at $5 \times 10^4$ cells/plate after TM treatment and cultured with EC growth medium for 72 h. Non-treated ECs were cultured in parallel as controls. At 24-, 48-, and 72-h time points, cell numbers from three dishes were counted to monitor growth rate. The cells were also stained with Ki-67 antibody (Cat. no. 0712020; Cell Signaling Technology) at the 24 h time point to monitor cell proliferation. For migration assays, the TM-treated and non-treated ECs were plated to 3-cm plates and cultured for 24 h, followed by removing attached cells growing on the bottom of the culture dishes with a cell scraper, leaving a 500 µm-wide area without cells. The dishes were continuously cultured for 24–30 h, followed by 4% formalin fixation and eosin staining of the cells. Cells that migrated to the gap areas were imaged under a microscope. The areas covered by migrated cells were measured, and the ratio of these areas to the original gap areas was calculated as the cell migration rate.

## Tube formation assays

Tube formation assays were performed following standard methods (51) with modification. Briefly, Matrigel Matrix (Cat. no. 354248; Corning) at 10 mg/ml concentration was used to coat a 24-well plate (0.3 ml/well) and a 96-well plate (50 $\mu$l/well). Cells were seeded to the Matrigel at the concentration of $4 \times 10^5$ cells/ml. After 6–12 h of culture without stimulation, the tube-like structures were recorded under the microscope. The total tube length was measured using ImageJ software.

## High throughput RNA sequencing and gene expression analysis

Two inducible ADAR1 KO mice from two independent litters were used for EC isolation. RNA samples were isolated from TM-treated and non-treated control cells, and RNA libraries were prepared

---

d-KO mice were analyzed for ISG expression. Compared to the EC-KO mice, ISG expression was significantly decreased in d-KO mice as shown by the seven representative ISGs. *P* < 0.05, n = 3 (WT) and 3–6 (i-KO and d-KO). **(C)** ISG expression was also tested in i-d-KO ECs after the *ADAR1* gene was deleted; ISG levels were significantly lower than the controls. *P* < 0.05, n = 3 (WT) and 3–4 (i-KO and d-KO). **(D)**. *ADAR1* and *MDA-5* gene deletion was monitored in the tested ECs with TM induction. *ADAR1* gene was efficiently deleted in both i-KO and i-d-KO ECs after TM induction, whereas the *MDA-5* gene was deleted in i-d-KO ECs with and without TM induction.

from 100 ng RNA using the KAPA RNA HyperPrep Kit with RiboErase (Kapa Biosystems) according to the manufacturer's protocol and sequenced using NovaSeq6000 platform (Illumina) to an average of 100M 101PE reads. The sequencing reads for each library (EC1 CON, EC1 TM, EC2 CON, and EC2 TM) were first quality-controlled using the FastQC tool (Andrews, Simon. "FastQC: a quality control tool for high throughput sequence data." 2010. https://www.bioinformatics.babraham.ac.uk/projects/fastqc/). Illumina adapter sequences and low-quality reads were filtered using the tool Trimmomatic (52). Then, surviving reads were aligned to the mouse reference genome mm10 by STAR aligner (53). Default parameter settings were applied to all pipelines of our analysis.

Based on the aligned RNA-seq data, gene counts were quantified using the tool HTSeq (54) for each library. On the top of the gene expression profiles across the four libraries, differential expression analysis was performed with R package DESeq2 (55). Differentially expressed genes (DEGs) were defined by FDR ≤ 5%, and DEGs with changes larger than 1.5-fold were further applied into Ingenuity Pathway Analysis (https://www.qiagenbioinformatics.com/products/ingenuitypathway-analysis) to detect pathways enriched by the gene alterations. Significant pathways were defined by FDR ≤ 5%.

### RNA editing site identification and analysis of RNA-seq data

For each RNA-seq library, variations were called based on the aligned file by the BCFtools mpileup function (56) and then annotated to known gene regions. Candidate editing sites were selected from these variation sites: (1) sites located at positive-strand gene regions and contained A-to-G mutation; or (2) sites located at negative-strand gene regions and contained T-to-C mutation at the reference strand. Based on these candidate editing sites, more stringent criteria were further applied to define more confident editing sites: (1) number of total reads covering the site ≥5; (2) number of altered reads covering the site ≥3; (3) editing rate (number of editing reads/number of total reads covering the site) ≤95%; and (4) editing rate ≥5%. These criteria were chosen to eliminate variations caused by rare sequencing error events but sacrifice rare real RNA editing events. These confident editing sites were annotated to different gene regions: 5′ distal (within 50 Kbp upstream of the gene starting point), promoter (within 1.0 Kbp upstream of the gene starting point), 5′ UTR, exon, intron, 3′ UTR, and downstream (within 1k bp downstream of the gene end point) and 3′ distal regions (within 50 Kbp downstream of the gene end point). In addition, RepeatMasker library was downloaded (Smit AFA, Hubley R, Green P (2013–2015) RepeatMasker Open-4.0. http://www.repeatmasker.org) and used to annotate the editing sites to repeat and non-repeat regions. Repetitive regions were categorized as SINE, long terminal repeat (LTR), long interspersed nuclear element (LINE), simple repeat, DNA, RNA, low complexity, and some other subtypes, where SINE can be further clustered as B1, B2, B4, and other subcategories.

To reduce the influence of single nucleotide polymorphisms (SNPs), libraries of the same mouse with control (CON) and TM treatment were compared. Editing sites of CON > TM (editing rate in control greater than TM-treated) were defined using the following criteria: (1) editing rate of CON–editing rate of TM ≥ 0.05; (2) number of total reads covering the CON variation site ≥5; (3) number of altered

reads covering the CON variation site ≥3; and (4) editing rate of CON ≤ 95%. Similarly, editing sites of TM > CON (editing rate in TM-treated greater than control) were defined as (1) editing rate of TM–editing rate of CON ≥ 0.05; (2) number of total reads covering the TM variation site ≥ 5; (3) number of edited/altered reads covering the TM variation site ≥3; and (4) editing rate of TM ≤ 95%. Consistent CON > TM (or CON < TM) sites across the two mice were further compared. Gene and repeat region distribution of these selected sites were explored.

In addition to editing site analysis, the data were further explored at the repetitive region (RR) level. RepeatMasker library was used to annotate the editing sites of each individual RNA-seq library or differential editing sites, comparing CON and TM libraries. Editing sites annotated to the same RR were grouped together. Editing rate for an RR was calculated by the average editing rates across all the "for-sure" editing sites located in that RR. Among all the RRs, we specifically focused on and explored SINE, LTR, and LINE.

### Data availability

All bulk RNA sequencing data (EC1 CON, EC1 TM, EC2 CON, and EC2 TM) were deposited into the NCBI Gene Expression Omnibus with an accession ID GSE174216. Raw read files, gene count quantification, and editing calling can be accessed via the link https://www.ncbi.nlm.nih.gov/geo/query/acc.cgi?acc=GSE174216.

### Statistics

Continuous data were summarized using mean and SD or median and interquartile range. Groups and categorical data were summarized using frequency and percentages. Bar, dot, and line graphs were used to depict differences and relationships in data. Wilcoxon rank sum test was used to test differences between two groups of nonparametric independent data. For differences between more than two groups of not normally distributed data, the nonparametric, Kruskal–Wallis test by ranks was used. Upon the rejecting the null of Kruskal–Wallis test, the post hoc Conover test was used for pairwise multiple comparisons procedure. All tests used in the analysis were of two-sided nature with $P \leq 0.05$ defined as statistically significant. Prism—GraphPad was used to generate publication quality graphs and Stata software version 12.0 (StataCorp) was used to conduct statistical testing. All statistical analyses on RNA-seq data and data visualizations were performed using R programming with the available R/Bioconductor packages.

# Supplementary Information

# Acknowledgements

We thank Christine Burr for editing the manuscript. Thanks to Dr. Stephen Chan for his comments on the manuscript. This study was supported by NIH Grant R01AI139544 and VA grant I01RX001455, was supported in part by the

NIH Center Core Grant, 1P30DK120531, and was partially supported by the UPMC Genome Center with funding from UPMC's Immunotherapy and Transplant Center.

## Author Contributions

X Guo: data curation, formal analysis, validation, investigation, and methodology.

S Liu: data curation, software, formal analysis, validation, visualization, methodology, and writing—original draft.

R Yan: data curation, investigation, and methodology.

V Nguyen: data curation, investigation, and methodology.

M Zenati: software and formal analysis.

TR Billiar: conceptualization, supervision, and writing—original draft, review, and editing.

Q Wang: conceptualization, resources, formal analysis, supervision, funding acquisition, validation, investigation, project administration, and writing—original draft, review, and editing.

## Conflict of Interest Statement

The authors declare that they have no conflict of interest.

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
