## [Reviewer comments · Life Science Alliance]

Life Science Alliance

ADAR1 RNA editing regulates endothelial cell functions via the MDA-5 RNA sensing signaling pathway

Xinfeng Guo, Silvia Liu, Rose Yan, Vy Nguyen, Mazen Zennatai, Timothy Billiar, and Qingde Wang

DOI: <https://doi.org/10.26508/lsa.202101191>

Corresponding author(s): Qingde Wang, University of Pittsburgh and Timothy Billiar, University of Pittsburgh

Review Timeline:

Submission Date:	2021-08-12
Editorial Decision:	2021-09-07
Revision Received:	2021-11-18
Editorial Decision:	2021-12-06
Revision Received:	2021-12-11
Accepted:	2021-12-13

Transaction Report:

September 7, 2021

Re: Life Science Alliance manuscript #LSA-2021-01191-T

Qingde Wang
University of Pittsburgh
Surgery
200 Lothrop St. BST W943
BSTW939
Pittsburgh, Pennsylvania 15213

Dear Dr. Wang,

Thank you for submitting your manuscript entitled "RNA sensing signaling pathway regulated by ADAR1 RNA editing modulates endothelial cell functions in neonatal mice" to Life Science Alliance. The manuscript was assessed by expert reviewers, whose comments are appended to this letter. We invite you to submit a revised manuscript addressing the Reviewer comments.

Thank you for this interesting contribution to Life Science Alliance. We are looking forward to receiving your revised manuscript.

Sincerely,

B. MANUSCRIPT ORGANIZATION AND FORMATTING:

Reviewer #1 (Comments to the Authors (Required)):

Guo et al describe their results on the role of ADAR1-mediated RNA editing in endothelial cells by investigating the effects of endothelial specific ADAR1 depletion.

The presented results are very interesting and focus on endothelial specific ADAR1-mediated effects. ADAR1 depletion decreased the editing levels on SINE RNAs in ECs, damaged multiple organs, activated the innate immune response and elevated the expression of ISGs via the MDA5 receptor, as shown by rescue experiments. The methods applied are appropriate. Yet, the description of the experimental designs is not always clear and also the discussion should focus on the results shown here and carefully discuss them in the context with the knowledge and results already described by others. This manuscript would profit from being written with more care and focus.

In Fig. 1A description of the genotypes used in this study is shown. To me, the breeding scheme remains unclear. What is the genotype of the breeding used in this study? Is it homo- or heterozygous for endothelial-specific ADAR1 deletion? Accordingly: are the 75% of early dying embryos homo- and heterozygous litters? While the surviving 25% are wildtype? ... or are these all homozygous or heterozygous litters with different expression intensities- as shown Figure 1G, in which the surviving population inactivated the cre-transgene.

Fig 1H shows the successful gene deletion in isolated ECs, but not in the tissue isolated from EC-ko mice. I cannot understand why no PCR signal is achieved in the highly vascularized tissue (as kidney, liver, brain, lung) examined here - and this is also not discussed. What is the age of the animals the tissue is isolated from ?

In Fig. 2A the lung is shown isolated from wildtype and EC-ko mice. In the Figure is shown that the right lung collapsed. This is also explicitly written in the text body. Is it in fact always the right lung and do the authors have any idea why this is so - this should be discussed.

In the text body (p7 line10) the authors write: „Pleural fluid often presented". I do not understand what is meant by this. The authors further show very interesting histological stainings and describe the CD31-immunohistological stainings of kidney and liver vessels as „enlarged tubular structures, confirming these were dilated or enlarged vessels". This finding is also not further discussed - neither in the context of inflammation nor in the context of ADAR1-/- induced apoptosis nor in the context of the results shown in Fig. 5, in which endothelial specific abilities were examined with respect to their proliferation, migration and tube-forming abilities.

Fig. 3B diagram: „IGS 15" - correct x-axis: should probably mean "ISG 15".

In the text body that belongs to Fig. 3, the authors write that they "again" turned to the ADAR1 i-KO ECs. This line is not described before and the order of the experiments has presumably been changed without changing the respective text body accordingly. The appropriate description of the cell line is then given in section „4. Cellular functions of endothelial cells are impaired".

Fig. 4 gives the RNA results for non-treated and TM treated ECs and shows that the interferon signaling pathway is largely upregulated in ADAR1-/- ECs. This is in accordance with data published previously for other cells and tissues and this should be discussed. The discussion misses a clear cut statement to what is specific for endothelial cell ADAR1-mediated signaling in comparison to other cell lines and tissues.

Also the MDA rescue experiments should be discussed on the background that ADAR1 signaling regulates the dsRNA sensing mechanism mediated by MDA5, MAVS, and MDA5-MAVS-IFN signaling (Liddicoat BJ et al. Science 2015, Mannion, N.M. et al. Cell Rep. 2014, Pestal-K et al. Immunity 2015). Accordingly it would be of interest whether the rescue is thought to work via MDA5 or the MDA5-MAVS-IFN signaling (see also Hartner JC et al. JBC 2004, Wang Q et al. JBC 2004).

In Supplemental Figure S1: the control staining is missing

Reviewer #2 (Comments to the Authors (Required)):

Review of Xingfeng Gu et al.,

In this manuscript the authors investigate the impact of a deletion of the RNA editing enzyme ADAR1 in endothelial cells. Specifically, the authors use a deletion allele of exons 11 through 15. The allele is floxed with the help of a caherin5-driven cre line.

The authors show that deletion of the C-terminus of ADAR1 leads to post natal death, which is in contrast to a total deletion of ADAR1 that normally leads to embryonic death. The authors provide evidence that endothelial cells are damaged leading to collapsed lungs but also to damage in other organs such as liver or kidney.

The authors show further that endothelial cells are impaired in tube formation and migration when ADAR1 is floxed out.

Gene expression analysis shows that interferon induced genes are strongly upregulated. The authors show further that critical editing in SINES is strongly reduced upon ADAR1 deletion most likely giving rise to the elevated interferon signaling.

Lastly, the authors show that an MDA5 deletion can rescue the observed lethality.

The manuscript is well written and the experiments are clearly documented and support the claims of the manuscript.
Major:

a) The fact that an endothelial cell deletion of ADAR can also be rescued by an MDA5 deletion is interesting as it indicates that the recoding function of ADAR1 is minor if any for endothelial function. Still, given the fact that the floxing of the ADAR1 allele is seemingly only partial, this point should be better addressed:

- 1) The editing patterns in DCSS should be shown in the presence and absence of ADAR1 and also in the dKO RNA
- 2) The cell migration and tube formation assay shown in figure 5 should also be repeated with the MDA-5 double KO, to verify that also this phenotype is dependent on the MDA-5 sensing pathway.

b) The ADAR1 allele used here should be better characterized. As the author will be well aware, there are large discrepancies in the ability of MDA-5 to rescue ADAR1 alleles. While the catalytic dead point mutation generated in the Walkley lab, can be fully rescued, a complete ADAR1 deletion available in the O'Connell lab cannot be rescued by MDA-5. Similarly, the deletions of Exons 7-9 can lead to a partial rescue upon crossing with MDA-5 or MAVS. Thus, it would be important to see how the deletion of exons 11-15 can be rescued by MDA5 (full body deletion). This is important, as it will allow the reader to assess whether the phenotype observed is due to editing deficiency, or RNA binding-deficiency.

c) Along the same lines, it would be important to see whether there is any residual ADAR protein being made in endothelial cells (western blot, RT-PCR).

Minor

ADAR1 i-KO should be explained in the text

Figure 2 should more clearly indicate the histological details by different symbols. The average molecular biologist cannot appreciate the details seen here.

We thank reviewers and editors for their careful evaluation of our manuscript. In the reversed version, we tried to incorporate all comments and suggestions, which made our manuscript significantly improved. Here is point to point answers to reviewers' concerns.

Answers for Reviewer #1:

Guo et al describe their results on the role of ADAR1-mediated RNA editing in endothelial cells by investigating the effects of endothelial specific ADAR1 depletion.

The presented results are very interesting and focus on endothelial specific ADAR1-mediated effects. ADAR1 depletion decreased the editing levels on SINE RNAs in ECs, damaged multiple organs, activated the innate immune Answer and elevated the expression of ISGs via the MDA5 receptor, as shown by rescue experiments. The methods applied are appropriate. Yet, the description of the experimental designs is not always clear and also the discussion should focus on the results shown here and carefully discuss them in the context with the knowledge and results already described by others. This manuscript would profit from being written with more care and focus.

Answer: We thank this reviewer for recognizing the significance of our work to determine the specific function of ADAR1 in endothelial cells and the underlying mechanism. We revised the description of the experimental design and added significant portion of discussions of our results according to the suggestions, on page 17-19.

In Fig. 1A description of the genotypes used in this study is shown. To me, the breeding scheme remains unclear. What is the genotype of the breeding used in this study? Is it homo- or heterozygous for endothelial-specific ADAR1 deletion? Accordingly: are the 75% of early dying embryos homo- and heterozygous litters? While the surviving 25% are wildtype? ... or are these all homozygous or heterozygous litters with different expression intensities- as shown Figure 1G, in which the surviving population inactivated the cre-transgene.

Answer: We added a detailed description of the breeding scheme on page 5, which was highlighted in yellow, lines 99-105. The breeding scheme is: ADAR1 wt/Lox; Cre+ genotype breed with ADAR1 Lox/Lox; Cre-, and in the progenies, the ADAR1 Lox/Lox; Cdh5-Cre+ was selected as the EC-specific ADAR1 KO mice for analysis. Both the 75% dying pups and the 25% survived mice were homozygous of ADAR1, and their genotype was ADAR1 Lox/Lox; Cdh5-Cre+. The Cre was hemizygous (one of the parents was Cre negative).

Yes, the survival was associated with the inactivation of the Cre transgene. The genotypes of the mice were specified each time they were described on pages 5-7.

Fig 1H shows the successful gene deletion in isolated ECs, but not in the tissue isolated from EC-ko mice. I cannot understand why no PCR signal is achieved in the highly vascularized tissue (as kidney, liver, brain, lung) examined here - and this is also not discussed. What is the age of the animals the tissue is isolated from ?

Answer: We performed additional experiments to find why the highly vascularized tissues did not show the PCR signal of ADAR1 gene deletion. For a semi-quantification of the relative amount of floxed and deleted ADAR1 genes, we used a mixture of 3 primers in our PCR reactions (P2 and P3 are antisense primers specific for floxed and deleted ADAR1 genes, respectively, and the two PCR amplicons share the sense primer P1, shown in Figure 1a), as we described previously, this PCR condition gives a relative quantification of the deleted gene refers to the undeleted floxed alleles (Am J Pathol. 2015,185:3224-37). However, the deleted gene signal was not well picked up when the deleted ADAR1 gene was less than 10%, in the PCR reaction with 3 mixed primers. In the EC-KO mouse model, the Cre recombination efficiency in ECs was not as high as we expected. In the tissue DNA samples, DNAs from other cell types further diluted the deleted ADAR1 alleles from the ECs, and thereby, the amplification was dominated by the undeleted floxed alleles. That was why the deleted signal was not seen. We performed new experiments to detect the deleted alleles in the tissue DNA samples by PCR with only the primers for deleted allele (Primer 1 and 3, without the primer P2), we observed the deleted gene signals. We added this result as supplemental Figure S7, and explained it in the corresponding legend and on Page 7 Line 137-139 and on Page 17, Line 365-368.

For Figure 1H, the tissues were from two weeks old pups, specified on Line 135-136 on page 7.

In Fig. 2A the lung is shown isolated from wildtype and EC-ko mice. In the Figure is shown that the right lung collapsed. This is also explicitly written in the text body. Is it in fact always the right lung and do the authors have any idea why this is so - this should be discussed.

Answer: The lung collapses did not always happen in the right lung. As shown in Figure 2A, the left lung also collapsed at the top lobe. We add an explanation to page 7, Line 145-146, highlighted in yellow.

We discussed this phenotype on page 17, Line 374-381.

In the text body (p7 line10) the authors write: "Pleural fluid often presented". I do not understand what is meant by this.

Answer: We revised this sentence. We often observed an increased amount of serous pleural fluid accumulated in the pleural cavities of mice with lung collapses, which did not present in the control mice. We highlighted the description on page 7, Line 148-149.

The authors further show very interesting histological staining and describe the CD31-immunohistological staining of kidney and liver vessels as "enlarged tubular structures, confirming these were dilated or enlarged vessels". This finding is also not further discussed - neither in the context of inflammation nor in the context of ADAR1-/- induced apoptosis nor in the context of the results shown in Fig. 5, in which endothelial specific abilities were examined with respect to their proliferation, migration and tube-forming abilities.

Answer: When we examined the microscopic structures of the kidney and liver on the HE stained sections, we saw the large empty spaces within the tissues without knowing what structures they were. When we stained the sections with CD31, we found the CD31 stained cells lining the spaces that

confirmed the spaces were enlarged vessels. We revised the description on page 8, Line 159-163, and discussed this phenotype on Page 17, Line 375-381.

Fig. 3B diagram: "IGS 15" - correct x-axis: should probably mean "ISG 15".

Answer: This was our mistake, and we changed it to "ISG 15".

In the text body that belongs to Fig. 3, the authors write that they "again" turned to the ADAR1 i-KO ECs. This line is not described before and the order of the experiments has presumably been changed without changing the respective text body accordingly. The appropriate description of the cell line is then given in section „4. Cellular functions of endothelial cells are impaired".

Answer: Thanks for pointing it out. We should have changed the description corresponding to the figures, and we have revised the description of the i-KO ECs in the text for Fig 3B, on Page 8, Line 175 to Page 9, line 181.

Fig. 4 gives the RNA results for non-treated and TM treated ECs and shows that the interferon signaling pathway is largely upregulated in ADAR-/- ECs. This is in accordance with data published previously for other cells and tissues and this should be discussed. The discussion misses a clear cut statement to what is specific for endothelial cell ADAR1-mediated signaling in comparison to other cell lines and tissues.

Answer: We added a paragraph to discuss the ISG expression in ADAR1 deficient ECs, and compared it to that described previously, on Page 18, Line 387-397.

Also the MDA rescue experiments should be discussed on the background that ADAR1 signaling regulates the dsRNA sensing mechanism mediated by MDA5, MAVS, and MDA5-MAVS-IFN signaling (Liddicoat BJ et al. Science 2015, Mannion, N.M. et al. Cell Rep. 2014, Pestal-K et al. Immunity 2015). Accordingly it would be of interest whether the rescue is thought to work via MDA5 or the MDA5-MAVS-IFN signaling (see also Hartner JC et al. JBC 2004, Wang Q et al. JBC 2004).

Answer: It is a very interesting question why MDA5 deletion rescued the ADAR1 deficient mice differently. While the Adar1^{E861A/E861A} mutant mice used by Liddicoat BJ et al. Science 2015 (knock-in of a single nucleotide mutation) was fully rescued, the deletion of MDA-5 or MAVS only partially rescued the embryos of the Adar1^{Δ7-9} knockout mice used by Pestal-K et al. Immunity 2015, the Adar1^{Δ2-13} used by Mannion, N. M. et al. Cell Rep. 2014, and the ADAR1 P150-/- mice used by Pestal-K et al. Immunity 2015, while these mice were not rescued to survive postnatally. In contrast, the Adar1^{Δ12-15} knockout in ECs in this study were completely rescued. We discussed this difference on Page 18, Line 399-426.

In Supplemental Figure S1: the control staining is missing

Answer: We added the control staining to figure S1 without the CD31 antibody. We also add the staining of the lung cells before EC purification. Before EC purification, many other cell types were not stained by CD31 antibodies. Very pure ECs were isolated for our analysis.

Co-author list 4Division of Viral Products - the author is missing

Answer: We mistakenly included this sentence, and it was removed.

Answers for Reviewer #2

Review of Xingfeng Gu et al.,

In this manuscript the authors investigate the impact of a deletion of the RNA editing enzyme ADAR1 in endothelial cells. Specifically, the authors use a deletion allele of exons 11 through 15. The allele is floxed with the help of a caherin5-driven cre line.

The authors show that deletion of the C-terminus of ADAR1 leads to post natal death, which is in contrast to a total deletion of ADAR1 that normally leads to embryonic death. The authors provide evidence that endothelial cells are damaged leading to collapsed lungs but also to damage in other organs such as liver or kidney.

The authors show further that endothelial cells are impaired in tube formation and migration when ADAR1 is floxed out.

Gene expression analysis shows that interferon induced genes are strongly upregulated. The authors show further that critical editing in SINES is strongly reduced upon ADAR1 deletion most likely giving rise to the elevated interferon signaling.

Lastly, the authors show that an MDA5 deletion can rescue the observed lethality.

The manuscript is well written and the experiments are clearly documented and support the claims of the manuscript.

Answer: We appreciate this reviewer's comments on our work. The major and minor concerns were addressed below.

Major:

a) The fact that an endothelial cell deletion of ADAR can also be rescued by an MDA5 deletion is interesting as it indicates that the recoding function of ADAR1 is minor if any for endothelial function. Still, given the fact that the floxing of the ADAR1 allele is seemingly only partial, this point should be better addressed:

Answer: We emphasized the partial deletion on Page 6, Line 119-123, highlighted in yellow. We also discussed the low Cre efficiency on Page 16, Line 361-363.

We agree with this reviewer on that the recoding function of ADAR1 is minor, although many studies in cardiologic field showed the significance of the recoding of specific RNAs. The regulation of RNA sensing seems to be a critical mechanism of ADAR1 in ECs.

1) The editing patterns in CTSS should be shown in the presence and absence of ADAR1 and also in the dKO RNA

Answer: The mRNA of cathepsin S (CTSS) was reported to play a critical role in human ECs. Nevertheless, we did not find a corresponding sequence in the mouse gene, nor a specific editing site in mouse CTSS mRNA. Our data did not support that a specific gene recoding plays a key role in ECs. We discussed it on Page 19-20, Line 428-441..

2) The cell migration and tube formation assay shown in figure 5 should also be repeated with the MDA-5 double KO, to verify that also this phenotype is dependent on the MDA-5 sensing pathway.

Answer: We added additional experimental data to address this question. The EC functions as tested by the in vitro assays were rescued, but it was not returned to the levels of the wild type controls. A minor MDA-5 independent function likely exists, but loss of this minor function can be tolerated by the rescued mice, which did not affect the mouse survival, and did not give an obvious phenotype. We added these results as Figure S9.

b) The ADAR1 allele used here should be better characterized. As the author will be well aware, there are large discrepancies in the ability of MDA-5 to rescue ADAR1 alleles. While the catalytic dead point mutation generated in the Walkley lab, can be fully rescued, a complete ADAR1 deletion available in the O'Connell lab cannot be rescued by MDA-5. Similarly, the deletions of Exons 7-9 can lead to a partial rescue upon crossing with MDA-5 or MAVS. Thus, it would be important to see how the deletion of exons 11-15 can be rescued by MDA5 (full body deletion). This is important, as it will allow the reader to assess whether the phenotype observed is due to editing deficiency, or RNA binding-deficiency.

Answer: This is a fascinating question. While the catalytic dead point mutation can be completely rescued by MDA-5, ADAR1 knockout mice, either deletion of E2-13 (coding Z-binding domain, dsRNA binding motif and catalytic domain) or E7-9 (coding dsRNA binding motif-3, and partial catalytic domain), can only be partially rescued by either MDA-5 or MAVS, and the mice could only survive up to one or a few days after birth. It seems that the RNA editing activity of ADAR1 prevents the activation of the MDA-5/MAVS pathway to rescue the embryos, and the functions of RNA binding/Z-binding functions have

other functions independent of the MDA-5/MAVS pathway. However, in contrast to the E2-13 and E7-9 knockout mice, we found that the postnatal lethality of EC-specific deletion of ADAR1 E12-15 can be completely rescued by MDA-5 deletion. We previously tested that the deletion of ADAR1 E12-15 did not result in truncated protein in the embryos, likely due to the instability of the truncated protein (JBC 2004, 279:4952; PNAS 2011, 108(24):E199). To determine whether the deletion of ADAR1 E12-15 preserved the N terminus of ADAR1 protein in ECs, we performed western blot on EC protein. The ADAR1 McAb, clone 15.8.6 against the dsRNA region, did not detect truncated ADAR1 in ECs. In addition, we bred the germline ADAR1 E12-15 KO mice with MDA-5 KO mice, the double knockout rescued the embryos to birth, but the pups died a few days later after birth (data not shown), the same as observed in E7-9 or E2-13 deleted mice. Therefore, the rescue of ADAR1 EC-KO mice from postnatal lethality by MDA-5 deletion was likely due to the EC function related, rather than preservation of partial ADAR1 protein. We discussed the different rescue effects of MDA-5 on page 18, Line 399-426.

We agree that this is an important question. Evidence from this study supports that the major function of ADAR1 in EC is to prevent MDA-5 mediated innate immune response. However, our model did not dissect the functions of different domains of ADAR1.

c) Along the same lines, it would be important to see whether there is any residual ADAR protein being made in endothelial cells (western blot, RT-PCR).

Answer: In this study, we choose to use Cdh5-Cre for our animal model preparation because of its high specificity for ECs. However, the efficiency of ADAR1 gene recombination is not very high. Our PCR analysis of the purified lung ECs showed that only about half or less of ADAR1 alleles were deleted. We do not find a method to separate the ADAR1 deleted ECs from ADAR1 undeleted cells isolated from the EC-KO mice for protein analysis. However, in the ECs isolated from the i-KO mice, the same floxed E12-15 of ADAR1 gene can be efficiently deleted. We performed a Western blot analysis on the i-KO EC protein samples. Both P150 and P110, or a truncated ADAR1 protein were not produced from the deleted ADAR1 gene. We added this result as **Figure S8**.

Minor

ADAR1 i-KO should be explained in the text

Answer: We have added description of the i-KO ECs in the text for Fig 3B, on Page 8, Line 1175 to Page 9, Line-181.

Figure 2 should more clearly indicate the histological details by different symbols. The average molecular biologist cannot appreciate the details seen here.

Answer: Additional description of pathologic feature was added to Fig 2, and discussed on Page 17, Line 375-381. Thanks for the suggestion.

December 6, 2021

RE: Life Science Alliance Manuscript #LSA-2021-01191-TR

Dr. Qingde Wang
University of Pittsburgh
Surgery
200 Lothrop St. BST W943
BSTW939
Pittsburgh, Pennsylvania 15213

Dear Dr. Wang,

Thank you for submitting your revised manuscript entitled "ADAR1 RNA editing regulates endothelial cell functions via the MDA-5 RNA sensing signaling pathway". We would be happy to publish your paper in Life Science Alliance pending final revisions necessary to meet our formatting guidelines.

- please upload your main and supplementary figures as single files; i.e. upload all figure files as individual ones, including the supplementary figure files (provide only one file for each figure)
- please add ORCID ID for secondary corresponding-he should have received instructions on how to do so
- please add the Twitter handle of your host institute/organization as well as your own or/and one of the authors in our system
- please note that titles in the system and manuscript file must match
- please add an Author Contributions section to your main manuscript text
- please add a conflict of interest statement to your main manuscript text
- we encourage you to revise the figure legends for figure 6 such that the figure panels are introduced in alphabetical order
- please use capital letters in the callouts for figures in the manuscript text
- please add callouts for Figures 1A-D; S3A-L; S4A-L; S5A-L; S6A-C; S9A-C to your main manuscript text

A. FINAL FILES:

B. MANUSCRIPT ORGANIZATION AND FORMATTING:

Sincerely,

Reviewer #1 (Comments to the Authors (Required)):

The manuscript sent by Guo et al about the role of ADAR1 in the RNA-sensing signaling pathway in endothelial cells is clear and concisely written. It gives very conclusive data on the impact of ADAR1 deletion on editing of cellular RNA, especially in RNA transcripts of SINEs. This deficit can be rescued by deletion of the MDA-5 receptor thereby blocking the RNA sensing signaling pathway without restoring editing deficits. The rescue effect is possibly based on the prevention of interferon-stimulated gene expression, the blocking of which is shown in this work.

Reviewer #2 (Comments to the Authors (Required)):

In this revised manuscript Go et al., take a thorough effort to address all criticisms raised during the review process. In particular questions to the completeness of the ADAR1 floxing, the type of the allele used and the production of residual proteins has been carefully addressed by additional experiments. Similarly, additional experiments were performed to demonstrate the impact of an Mda5 deletion on vascular outgrowth. Overall, the present manuscript has significantly improved and the authors have addressed all issues raised by this reviewer.

December 13, 2021

RE: Life Science Alliance Manuscript #LSA-2021-01191-TRR

Dr. Qingde Wang
University of Pittsburgh
Surgery
200 Lothrop St. BST W943
BSTW939
Pittsburgh, Pennsylvania 15213

Dear Dr. Wang,

Thank you for submitting your Research Article entitled "ADAR1 RNA editing regulates endothelial cell functions via the MDA-5 RNA sensing signaling pathway". It is a pleasure to let you know that your manuscript is now accepted for publication in Life Science Alliance. Congratulations on this interesting work.

DISTRIBUTION OF MATERIALS:

Again, congratulations on a very nice paper. I hope you found the review process to be constructive and are pleased with how the manuscript was handled editorially. We look forward to future exciting submissions from your lab.

Sincerely,
